# The Power of Comparisons for Actively Learning Linear Classifiers

**Max Hopkins**
Dept. of Computer Science and Engineering
University of California, San Diego
La Jolla, CA 92092
nmhopkin@eng.ucsd.edu

**Daniel Kane**
Dept. of Computer Science and Engineering
University of California, San Diego
La Jolla, CA 92092
dakane@eng.ucsd.edu

**Shachar Lovett**
Dept. of Computer Science and Engineering
University of California, San Diego
La Jolla, CA 92092
slovett@cs.ucsd.edu

## Abstract

In the world of big data, large but costly to label datasets dominate many fields. Active learning, a semi-supervised alternative to the standard PAC-learning model, was introduced to explore whether adaptive labeling could learn concepts with exponentially fewer labeled samples. While previous results show that active learning performs no better than its supervised alternative for important concept classes such as linear separators, we show that by adding weak distributional assumptions and allowing comparison queries, active learning requires exponentially fewer samples. Further, we show that these results hold as well for a stronger model of learning called Reliable and Probably Useful (RPU) learning. In this model, our learner is not allowed to make mistakes, but may instead answer "I don't know." While previous negative results showed this model to have intractably large sample complexity for label queries, we show that comparison queries make RPU-learning at worst logarithmically more expensive in both the passive and active regimes.

## 1 Introduction

In recent years, the availability of big data and the high cost of labeling has lead to a surge of interest in *active learning*, an adaptive, semi-supervised learning paradigm. In traditional active learning, given an instance space $X$, a distribution $D$ on $X$, and a class of concepts $c : X \to \{0, 1\}$, the learner receives unlabeled samples x from $D$ with the ability to query an oracle for the labeling $c(x)$. Classically our goal would be to minimize the number of samples the learner draws before approximately learning the concept class with high probability (PAC-learning). Instead, active learning assumes unlabeled samples are inexpensive, and rather aims to minimize expensive queries to the oracle. While active learning requires exponentially fewer labeled samples than PAC-learning for simple classes such as thresholds in one dimension, it fails to provide asymptotic improvement for classes essential to machine learning such as linear separators [1].

However, recent results point to the fact that with slight relaxations or additions to the paradigm, such concept classes can be learned with exponentially fewer queries. In 2013, Balcan and Long [2] proved that this was the case for homogeneous (through the origin) linear separators, as long as the distribution over the instance space $X = \mathbb{R}^d$ was (nearly isotropic) log-concave–a wide range of distributions generalizing common cases such as gaussians or uniform distributions over convex sets.

Later, Balcan and Zhang [3] extended this to isotropic s-concave distributions, a diverse generalization of log-concavity including fat-tailed distributions. Similarly, El-Yaniv and Wiener [4] proved that non-homogeneous linear separators can be learned with exponentially fewer queries over gaussian distributions with respect to the accuracy parameter, but require exponentially more queries in the dimension of the instance space $X$, making their algorithm intractable in high dimensions.

Kane, Lovett, Moran, and Zhang (KLMZ) [5] broke the non-homogeneity barrier for general distributions in two dimensions by empowering the oracle to compare points rather than just label them. Queries of this type are called comparisons, and are notable not only for their increase in computational power, but for their real world applications such as in recommender systems [6] or for increasing accuracy over purely label-based techniques [7]. Our work adopts a mixture of the approaches of Balcan, Long, and Zhang, and KLMZ. By leveraging comparison queries, we provide a computationally efficient algorithm for active learning non-homogeneous linear separators in exponentially fewer samples as long as the distribution satisfies weak concentration and anti-concentration bounds, conditions realized by, for instance, (not necessarily isotropic) s-concave distributions. Further, by introducing a new average case complexity measure, *average inference dimension*, that extends KLMZ's techniques to the distribution dependent setting, we prove that comparisons provide significantly stronger guarantees than the PAC-learning paradigm.

In the late 80's, Rivest and Sloan [8] proposed a competing model to PAC-learning called Reliable and Probably Useful (RPU) learning. This model, a learning theoretic formalization of Chow's [9] *selective classification*, does not allow the learner to make mistakes, but instead allows the answer "I don't know" written as "⊥". Here, error is measured not by the amount of misclassified examples, but by the measure of examples on which our learner returns ⊥. RPU-learning was for the most part abandoned by the early 90's in favor of PAC-learning as Kivinen [10–12] proved the sample complexity of RPU-learning simple concept classes such as rectangles required an exponential number of samples even under the uniform distribution. However, the model was recently re-introduced by El-Yaniv and Wiener [4], who termed it *perfect selective classification*. El-Yaniv and Wiener prove a connection between Active and RPU-learning similar to the strategy employed by KLMZ [5] (who refer to RPU-learners as "confident" learners). We extend the lower bound of El-Yaniv and Wiener to prove that actively RPU-learning linear separators with only labels is exponentially difficult in dimension even for nice distributions. On the other hand, through a structural analysis of average inference dimension, we prove that comparison queries allow RPU-learning with nearly matching sample and query complexity to PAC-learning, as long as the underlying distribution is sufficiently nice. This last result has already found further use by Hopkins, Kane, Lovett, and Mahajan [13], who use our analysis of average inference dimension to extend their comparison-based algorithms for robustly learning non-homogeneous hyperplanes to higher dimensions.

## 2 Background and related work

### 2.1 PAC-learning

Probably Approximately Correct (PAC)-learning is a framework for learning classifiers over an instance space introduced by Valiant [14] with aid from Vapnik and Chervonenkis [15]. Given an instance space $X$, label space $Y$, and a concept class $C$ of concepts $c : X \to Y$, PAC-learning proceeds as follows. First, an adversary chooses a hidden distribution $D$ over $X$ and a hidden classifier $c \in C$. The learner then draws labeled samples from $D$, and outputs a concept $c'$ which it thinks is close to $c$ with respect to $D$. Formally, we define closeness of $c$ and $c'$ as the error:

$$err_D(c', c) = Pr_{x \in D}[c'(x) \neq c(x)].$$

We say the pair $(X, C)$ is PAC-learnable if there exists a learner $A$ which, for all $\varepsilon, \delta > 0$, picks in $n(\varepsilon, \delta) = \text{Poly}(\frac{1}{\varepsilon}, \frac{1}{\delta})$ samples[1] a classifier $c'$ that with probability $1 - \delta$ has at most $\varepsilon$ error from $c$:

$$\exists A \ s.t. \ \forall c \in C, \forall D, Pr_{S \sim D^{n(\varepsilon, \delta)}}[err_D(A(S), c) < \varepsilon] \geq 1 - \delta.$$

The goal of PAC-learning is to compute the sample complexity $n(\varepsilon, \delta)$ and thereby prove whether certain pairs $(X, C)$ are efficiently learnable. In this paper, we will be concerned with the case of binary classification, where $Y = \{0, 1\}$. In addition, in the case that $C$ is linear separators we instead write our concept classes as the sign of a family $H$ of functions $h : X \to \mathbb{R}$. Instead of $(X, C)$,

we write the *hypothesis class* $(X, H)$, and each $h \in H$ defines a concept $c_h(x) = sgn(h(x))$. The sample complexity of PAC-learning is characterized by the VC dimension [16–18] of $(X, H)$ which we denote by $k$, and is given by:

$$n(\varepsilon, \delta) = \theta \left( \frac{k + \log(\frac{1}{\delta})}{\varepsilon} \right).$$

## 2.2 RPU-learning

Reliable and Probably Useful (RPU)-learning is a stronger variant of PAC-learning introduced by Rivest and Sloan [8], in which the learner is reliable: it is not allowed to make errors, but may instead say "I don't know" (or for shorthand, "$\perp$"). Since it is easy to make a reliable learner by simply always outputting "$\perp$", our learner must be useful, and may only output "$\perp$" a small fraction of the time. Let $A$ be a reliable learner, we define the error of $A$ on a sample S with respect to $D, c$ to be

$$err_D(A(S), c) = Pr_{x \sim D}[A(S)(x) = \perp].$$

We call $1 - err_D(A(S), c)$ the *coverage* of the learner $A$, denoted $C_D(A(S))$, or just $C(A)$ when clear from context. Finally, we say the pair $(X, C)$ is RPU-learnable if $\forall \varepsilon, \delta$, there exists a reliable learner $A$ which in $n(\varepsilon, \delta) = \text{Poly}(\frac{1}{\varepsilon}, \frac{1}{\delta})$ samples has error $\leq \varepsilon$ with probability $\geq 1 - \delta$:

$$\exists A \ s.t. \ \forall c \in C, \forall D, Pr_{S \sim D^{n(\varepsilon, \delta)}}[err_D(A(S), c) \leq \varepsilon] \geq 1 - \delta$$

RPU-learning is characterized by the VC dimension of certain intersections of concepts [11]. Unfortunately, many simple cases turn out to be not RPU-learnable (e.g. rectangles in $[0, 1]^d$, $d \geq 2$ [10]), with even relaxations having exponential sample complexity [12].

## 2.3 Passive vs active learning

PAC and RPU-learning traditionally refer to supervised learning, where the learning algorithm receives pre-labeled samples. We call this paradigm passive learning. In contrast, active learning refers to the case where the learner receives unlabeled samples and may adaptively query a labeling oracle. Similar to the passive case, for active learning we study the query complexity $q(\varepsilon, \delta)$, the minimum number of queries to learn some pair $(X, C)$ in either the PAC or RPU learning models. The hope is that by adaptively choosing when to query the oracle, the learner may only need to query a number of samples logarithmic in the sample complexity.

We will discuss two paradigms of active learning: pool-based, and membership query synthesis (MQS) [19, 20]. In the former, the learner has access to a pool of unlabeled data and may request that the oracle label any point. This model matches real-world scenarios where learners have access to large, unlabeled datasets, but labeling is too expensive to use passive learning (e.g. medical imagery). Membership query synthesis allows the learner to synthesize points in the instance space and query their labels. This model is the logical extreme of the pool-based model where our pool is the entire instance space. Because we consider learning with a fixed distribution, we will slightly modify MQS: the learner may only query points in the support of the distribution.[2] This is the natural specification to distribution dependent learning, as it still models the case where our pool is as large as possible.

## 2.4 The distribution dependent case

While PAC and RPU-learning were traditionally studied in the worst-case scenario over distributions, data in the real world is often drawn from distributions with nice properties such as concentration and anti-concentration bounds. As such, there has been a wealth of research into distribution-dependent PAC-learning, where the model has been relaxed only in that some distributional conditions are known. Distribution dependent learning has been studied in both the passive and the active case [2, 21–23]. Most closely related to our work, Balcan and Long [2] proved new upper bounds on active and passive learning of homogeneous (through the origin) linear separators in 0-centered log-concave distributions. Later, Balcan and Zhang [3] extended this to isotropic s-concave distributions. We directly extend the original algorithm of Balcan and Long to non-homogeneous linear separators via the inclusion of comparison queries, and leverage the concentration results of Balcan and Zhang to provide an inference based algorithm for learning under general s-concave distributions.

### 2.4.1 Point location

Our results on RPU-learning imply simple linear decision trees (LDTs) for an important problem in computer science and computational geometry known as point location. Given a set of $n$ hyperplanes in $d$ dimensions, called a *hyperplane arrangement* of size $n$ and denoted by $H = \{h_1, \ldots, h_n\}$, it is a classic result that $H$ partitions $\mathbb{R}^d$ into $O(n^d)$ cells. The point location problem is as follows:

**Definition 2.1** (Point Location Problem). *Given a hyperplane arrangement $H = \{h_1, \ldots, h_n\}$ and a point $x$, both in $\mathbb{R}^d$, determine in which cell of $H$ $x$ lies.*

Instances of this problem show up throughout computer science, such as in $k$-sum, subset-sum, knapsack, or any variety of other problems [24]. The best known depth for an LDT solving the point location problem is from a recent work of Hopkins, Kane, Lovett, and Mahajan [25], who prove the existence of a nearly optimal $\tilde{O}(d \log(n))$ depth LDT for arbitrary $H$ and $x$. The caveat of this work is that the LDT uses arbitrary linear queries, which may be too powerful a model in practice. Kane, Lovett, and Moran [26] offer an $\tilde{O}(d^4 \log(n))$ depth LDT restricting the model to generalized comparison queries, queries of the form $sgn(a\langle h_1, x \rangle - b\langle h_2, x \rangle)$ for a point $x$ and hyperplanes $h_1, h_2$. These queries are nice as they preserve structural properties of the input $H$ such as sparsity, but they still suffer from over-complication–any $H$ allows an infinite set of queries.

KLMZ's [5] original work on inference dimension showed that in the worst case, the depth of a comparison LDT for point location is $\Omega(n)$. However, by restricting $H$ to have good margin or bounded bit complexity, they build a comparison LDT of depth $\tilde{O}(d \log(n))$, which comes with the advantage of drawing from a finite set of queries for a given problem instance. Our work provides another result of this flavor: we will prove that if $H$ is drawn from a distribution with weak restrictions, for large enough $n$ there exists a comparison LDT with expected depth $\tilde{O}(d \log^2(n))$.

## 3  Results

We begin by introducing notation for our learning models. For a distribution $D$, an instance space $X \subseteq \mathbb{R}^d$, and a hypothesis class $H : X \to \mathbb{R}$, we write the triple $(D, X, H)$ to denote the problem of learning a hypothesis $h \in H$ with respect to $D$ over $X$. When $D$ is the uniform distribution over $S \subseteq X$, we will write $(S, X, H)$ for convenience. We will further denote by $B^d$ the unit ball in $d$ dimensions, and by $H_d$ hyperplanes in $d$ dimensions. Given $h \in H$ and a point $x \in X$, a *label query* determines $\text{sign}(h(x))$; given $x, x' \in X$, a *comparison query* determines $\text{sign}(h(x) - h(x'))$.

In addition, we will separate our models of learnability into combinations of three classes Q,R, and S, where Q $\in$ {Label, Comparison}, R $\in$ {Passive, Pool, MQS}, and S $\in$ {PAC, RPU}. Informally, we say an element $Q$ defines our query type, an element in $R$ our learning regime, and an element in $S$ our learning model. Learnability of a triple is then defined by the combination of any choice of query, regime, and model, which we term as the $Q$-$R$-$S$ learnability of $(D, X, H)$. Note that in Comparison-learning we have both a labeling and comparison oracle.

Finally, we will discuss a number of different measures of complexity for $Q$-$R$-$S$ learning triples. For passive learning, we will focus on the sample complexity $n(\varepsilon, \delta)$. For active learning, we will focus on the query complexity $q(\varepsilon, \delta)$. In both cases, we will often drop $\delta$ and instead give bounds on the *expected* sample/query complexity for error $\varepsilon$ denoted $\mathbb{E}[n(\varepsilon)]$ (or $q(\varepsilon)$ respectively), the expected number of samples/queries needed to reach $\varepsilon$ error. A bound for probability $1 - \delta$ then follow with $\log(1/\delta)$ repetitions by Chernoff. In the case of a finite instance space $X$ of size $n$, we denote the expected query complexity of perfectly learning $X$ as $\mathbb{E}[q(n)]$.

Finally, we use a subscript $d$ in our asymptotic notation to suppress factors dependent on dimension.

### 3.1  PAC-Learning

To show the power of active learning with comparison queries in the PAC-learning model, we will begin by proving lower bounds. In particular, we show that neither active learning nor comparison queries alone provide a significant speed-up over passive learning. In order to do this, we will assume the stronger MQS model, as lower bounds here transfer over to the pool-based regime.

**Proposition 3.1.** *For small enough $\varepsilon$, and $\delta = \frac{1}{2}$, the query complexity of Label-MQS-PAC learning $(B^d, \mathbb{R}^d, H_d)$ is:*

$$q(\varepsilon, 1/2) = \Omega_d\left(\left(\frac{1}{\varepsilon}\right)^{\frac{d-1}{d+1}}\right).$$

Thus without enriched queries, active learning fails to significantly improve over passive learning even over a nice distributions. Likewise, adding comparison queries alone also provides little improvement.

**Proposition 3.2.** *For small enough $\varepsilon$, and $\delta = \frac{3}{8}$, the sample complexity of Comparison-Passive-PAC learning $(B^d, \mathbb{R}^d, H_d)$ is:*

$$n(\varepsilon, 3/8) = \Omega\left(\frac{1}{\varepsilon}\right).$$

Now we can compare the query complexity of active learning with comparisons to the above. For our upper bound, we will assume the pool-based model with a $\mathrm{Poly}(1/\varepsilon, \log(1/\delta))$ pool size, as upper bounds here transfer to the MQS model. Our algorithm for Comparison-Pool-PAC learning combines a modification of Balcan and Long's [2] learning algorithm with noisy thresholding to provide an exponential speed-up for non-homogeneous linear separators.

**Theorem 3.3.** *Let $D$ be a log-concave distribution over $\mathbb{R}^d$. Then the query complexity of Comparison-Pool-PAC learning $(D, \mathbb{R}^d, H_d)$ is*

$$q(\varepsilon, \delta) = \tilde{O}\left(\left(d + \log\left(\frac{1}{\delta}\right)\right)\log\left(\frac{1}{\varepsilon}\right)\right).$$

Kulkarni, Mitter, and Tsitsiklis [27] (with analysis from [2]) also give a lower bound of $d\log(1/\varepsilon)$ for log-concave distributions for arbitrary binary queries, so Theorem 3.3 is near tight in dimension and error. It should be noted, however, that to cover non-isotropic distributions, Theorem 3.3 must know the exact distribution $D$. This restriction becomes unnecessary if $D$ is promised to be isotropic.

## 3.2 RPU-Learning

In the RPU-learning model, we will first confirm that passive learning with label queries is intractable information theoretically, and continue to show that active learning alone provides little improvement. Unlike in PAC-learning however, we will show that comparisons in this regime provide a significant improvement in not only active, but also passive learning.

**Proposition 3.4.** *The expected sample complexity of Label-Passive-RPU learning $(B^d, \mathbb{R}^d, H_d)$ is:*

$$\mathbb{E}[n(\varepsilon)] = \tilde{\Theta}_d\left(\left(\frac{1}{\varepsilon}\right)^{\frac{d+1}{2}}\right).$$

Thus we see that RPU-learning linear separators is intractable for large dimension. Further, active learning with label queries is of the same order of magnitude.

**Proposition 3.5.** *For all $\delta < 1$, the query complexity of Label-MQS-RPU learning $(B^d, \mathbb{R}^d, H_d)$ is:*

$$q(\varepsilon, \delta) = \Omega_d\left(\left(\frac{1}{\varepsilon}\right)^{\frac{d-1}{2}}\right).$$

In the appendix, we show that this bound is tight up to a logarithmic factor. For passive RPU-learning with comparison queries, we can inherit the lower bound from the PAC model (Proposition 3.2).

**Corollary 3.6.** *For small enough $\varepsilon$, and $\delta = \frac{3}{8}$, any algorithm that Comparison-Passive-RPU learns $(B^d, \mathbb{R}^d, H_d)$ must use at least*

$$n(\varepsilon, 3/8) = \Omega\left(\frac{1}{\varepsilon}\right)$$

*samples.*

Note that unlike the case of label queries, this lower bound is not exponential in dimension. In fact, not only is this bound tight up to a linear factor in dimension, comparison queries in general shift the RPU model from being intractable to losing only a logarithmic factor over PAC-learning in both the passive and active regimes. We need one definition: two distributions $D, D'$ over $\mathbb{R}^d$ are affinely equivalent if there is an invertible affine map $f : \mathbb{R}^d \to \mathbb{R}^d$ such that $D(x) = D'(f(x))$.

**Theorem 3.7.** *Let $D$ be a distribution over $\mathbb{R}^d$ that is affinely equivalent to a distribution $D'$ over $\mathbb{R}^d$, for which the following holds:*

*1. $\forall \alpha > 0$, $Pr_{x \sim D'}[\|x\| > d\alpha] \leq \frac{c_1}{\alpha}$*

*2. $\forall \alpha > 0$, $\langle v, \cdot \rangle + b \in H_d$, $Pr_{x \sim D'}[|\langle x, v \rangle + b| \leq \alpha] \leq c_2\alpha$*

*The sample complexity of Comparison-Passive-RPU-learning $(D, \mathbb{R}^d, H_d)$ is:*

$$n(\varepsilon, \delta) = \tilde{O}\left(\frac{d \log\left(\frac{1}{\delta}\right) \log\left(\frac{1}{\varepsilon}\right)}{\varepsilon}\right),$$

*and the query complexity of Comparison-Pool-RPU learning $(D, \mathbb{R}^d, H_d)$ is:*

$$q(\varepsilon, \delta) = \tilde{O}\left(d \log\left(\frac{1}{\delta}\right) \log^2\left(\frac{1}{\varepsilon}\right)\right).$$

*Note that the constants have logarithmic dependence on $c_1$ and $c_2$.*

Theorem 3.7 is not only computationally efficient, running in time $\text{Poly}(d, 1/\varepsilon, \log(1/\delta))$, but also applies to a wide range of distributions. It includes, for instance, the class of s-concave distributions for $s \geq -\frac{1}{2d+3}$ [3], notably removing the common requirement of isotropy [2, 28, 3].

We view Theorem 3.7 and its surrounding context as this work's main technically novel contribution. In particular, to prove the result, we introduce a new average-case complexity measure called average inference dimension that extends the theory of inference dimension from [5] (See Section 4.2). Further, this framework allows our analysis to extend to the point location problem as well.

**Theorem 3.8.** *Let $D$ be a distribution satisfying the criterion of Theorem 3.7, $x \in \mathbb{R}^d$, and $h_1, \ldots, h_n \sim D$. Then for large enough $n$ there exists an LDT using only label and comparison queries solving the point location problem with expected depth $\tilde{O}(d \log^2(n))$.*

For ease of viewing, we summarize our main results on expected sample/query complexity in Tables 1 and 2 for the special case of the uniform distribution over the unit ball. The bounds not novel to this work are the Label-Passive-PAC bounds [21, 18], and the lower bound on Comparison-Pool/MQS-PAC learning [2, 27]. Note also that lower bounds for PAC learning carry over to RPU learning.

Table 1: Expected sample and query complexity for PAC learning $(B^d, \mathbb{R}^d, H_d)$.

| PAC | Passive | Pool | MQS |
|---|---|---|---|
| Label | $\Theta\left(\frac{d}{\varepsilon}\right)$ [21, 18] | $\Omega_d\left(\left(\frac{1}{\varepsilon}\right)^{\frac{d-1}{d+1}}\right)$ | $\Omega_d\left(\left(\frac{1}{\varepsilon}\right)^{\frac{d-1}{d+1}}\right)$ |
| Comparison | $\Omega\left(\frac{1}{\varepsilon}\right)$ | $\tilde{\Theta}\left(d \log\left(\frac{1}{\varepsilon}\right)\right)$ | $\tilde{\Theta}\left(d \log\left(\frac{1}{\varepsilon}\right)\right)$ [2, 27] |

Table 2: Expected sample and query complexity for RPU learning $(B^d, \mathbb{R}^d, H_d)$.

| RPU | Passive | Pool | MQS |
|---|---|---|---|
| Label | $\tilde{\Theta}_d\left(\left(\frac{1}{\varepsilon}\right)^{\frac{d+1}{2}}\right)$ | $\tilde{\Omega}_d\left(\left(\frac{1}{\varepsilon}\right)^{\frac{d-1}{2}}\right)$ | $\tilde{\Omega}_d\left(\left(\frac{1}{\varepsilon}\right)^{\frac{d-1}{2}}\right)$ |
| Comparison | $\tilde{O}\left(\frac{d}{\varepsilon}\right)$ | $\tilde{O}\left(d \log^2\left(\frac{1}{\varepsilon}\right)\right)$ | $\tilde{O}\left(d \log^2\left(\frac{1}{\varepsilon}\right)\right)$ |

# 4 Techniques

## 4.1 Lower bounds: caps and polytopes

Our lower bounds for both the PAC and RPU models rely mainly on high-dimensional geometry. For PAC-learning, we consider spherical caps, portions of $B^d$ cut off by a hyperplane. Our two lower bounds, Label-MQS-PAC, and Comparison-Passive-PAC, consider different aspects of these objects. The former (Proposition 3.1) employs a packing argument: if an adversary chooses a hyperplane uniformly among a set defining some packing of (sufficiently large) caps, the learner is forced to query a point in many of them in order to distinguish which is labeled negatively. The latter bound (Proposition 3.2), follows from an indistinguishability argument: if an adversary chooses just between one hyperplane defining some (sufficiently large) cap, and the corresponding parallel hyperplane tangent to $B^d$, the learner must draw a point near the cap before it can distinguish between the two.

For RPU-learning, our lower bounds rely on the average and worst-case complexity of polytopes. For Label-Passive-RPU learning (Propositions 3.4), we consider random polytopes, convex hulls of samples $S \sim D^n$, whose complexity $E(D, n)$ is the expected probability mass across samples of size $n$. In this regime, we consider an adversary who, with high probability, picks a distribution in which almost all samples are entirely positive. As a result, the learner cannot infer any point outside of the convex hull of their sample, which bounds their expected coverage by $E(D, n)$. For Label-MQS-RPU learning (Proposition 3.5), the argument is much the same, substituting maximum probability mass for expectation. These techniques are generalizations of El-Yaniv and Wiener's [4] algorithm specific lower bounds, which also employ random polytope complexity.

## 4.2 Upper Bounds: Average Inference Dimension

We focus in this section on techniques used to prove our RPU-learning upper bounds, which we consider our most technically novel contribution. To prove our Comparison-Pool-RPU upper bound and corresponding point location result, Theorems 3.7 and 3.8, we introduce a novel extension to the inference dimension framework of KLMZ [5]. Inference dimension is a combinatorial complexity measure that characterizes the distribution independent query complexity of active learning with enriched queries. KLMZ show, for instance, that linear separators in $\mathbb{R}^2$ may be Comparison-Pool-PAC learned in only $\tilde{O}(\log(\frac{1}{\varepsilon}))$ queries, but require $\Omega\left(\frac{1}{\varepsilon}\right)$ queries in three or more dimensions.

Given a hypothesis class $(X, H)$, and a set of binary queries $Q$ (e.g. labels and comparisons), denote the answers to all queries on $S \subseteq X$ by $Q(S)$. Inference dimension examines the size of $S$ necessary to *infer* another point $x \in X$, where $S$ infers the point $x$ under $h$, denoted

$$S \to_h x,$$

if $Q(S)$ under $h$ determines the label of $x$. As an example, let $H$ be linear separators in $d$ dimensions, $Q$ be label queries, and our sample $d + 1$ points in convex position, positively labeled under some classifier $h$. Due to linearity, any point inside the convex hull of $S$ is inferred by $S$ under $h$.

Then in greater detail, the inference dimension of $(X, H)$ is the minimum $k$ such that in any subset of $X$ of size $k$, at least one point can be inferred from the rest:

**Definition 4.1** (Inference Dimension [5]). *The inference dimension of $(X, H)$ with query set $Q$ is the smallest $k$ such that for any subset $S \subset X$ of size $k$, $\forall h \in H$, $\exists x \in S$ s.t. $Q(S - \{x\})$ infers $x$ under $h$.*

KLMZ show that finite inference dimension implies distribution independent query complexity that is logarithmic in the sample complexity. On the other hand, they prove a lower bound showing that PAC learning classes with infinite inference dimension requires at least $\Omega(1/\varepsilon)$ queries.

To overcome this lower bound (which holds for linear separators in three plus dimensions), we introduce a distribution dependent version of inference dimension which examines the probability that a sample contains no point which can be inferred from the rest.

**Definition 4.2** (Average Inference Dimension). *We say $(D, X, H)$ has average inference dimension $g(n)$, if:*

$$\forall h \in H, Pr_{S \sim D^n}[\nexists x \text{ s.t. } S - \{x\} \to_h x] \leq g(n).$$

Small average inference dimension implies that finite samples have low inference dimension with high probability (Obs 4.6, Appendix). Theorems 3.7 and 3.8 then follow from our main technical contribution (Thm 4.10, Appendix): the average inference dimension of $(D, \mathbb{R}^d, H_d)$ with respect to comparisons is $2^{-\Omega_d(n^2)}$, so long as $D$ satisfies the weak distributional requirements of Theorem 3.7.

## 5 Experimental results

To confirm our theoretical findings, we have implemented a variant of Theorem 3.7 for finite samples. For a given sample size or dimension, the query complexity we present is averaged over 500 trials of the algorithm.

### 5.1 Algorithm

We first note a few practical modifications. First, our algorithm labels finite samples drawn from the uniform distribution over $B^d$. Second, to match our methodology in lower bounding Label-Pool-RPU learning, we will draw our classifier uniformly from hyperplanes tangent to the unit ball. Finally, because the true inference dimension of the sample might be small, our algorithm guesses a low potential inference dimension to start, and doubles its guess on each iteration with low coverage.

Our algorithm will reference two sub-routines employed by the original inference dimension algorithm in [5], Query$(Q, S)$, and Infer$(S, C)$. Query$(Q, S)$ simply returns $Q(S)$, the oracle responses to all queries on $S$ of type $Q$. Infer$(S, C)$ builds a linear program from constraints $C$ (solutions to some Query$(Q, S)$), and returns which points in $S$ are inferred.

---

**Algorithm 1:** Perfect-Learning$(N, Q, d, c)$

---

**Result:** Labels all points in sample $S \sim (B^d)^N$ using query set $Q$

1  S $\sim (B^d)^N$; Classifier $\sim S^d, B^1$;
2  Subsample_Size $= d + 1$; Uninferred $= S$; Subsample_List = [];
3  **while** *size(Uninferred) > c · size(Subsample_Size)* **do**
4  |    Subsample $\sim$ Uninferred[Subsample_Size];
5  |    Subsample_List.extend(Subsample);
6  |    Inferred_Points = Infer(Uninferred, Query(Q, Subsample_List));
7  |    **if** *size(Inferred_Points) < size(Uninferred)/2* **then**
8  |      | Subsample_Size $\ast= 2$;
9  |    **end**
10 |    Uninferred.remove(Inferred_Points)
11 **end**
12 Query(Label,Uninferred)

---

Algorithm 1 is efficient. The while loop runs at most $\log(N)$ times and solves at most $N$ linear programs with $O(f_Q(N))$ constraints in $d + 1$ variables, giving a total running time of Poly$(N, d)$.

### 5.2 Query complexity

Theorem 3.7 states that for an adversarial choice of classifier, the number of queries Perfect-Learning$(N, \text{Comparison}, d)$ performs is logarithmic compared to Perfect-Learning$(N, \text{Labels}, d)$. The left graph in Figure 1 shows this correspondence for uniformly drawn hyperplanes tangent to the unit ball and sample values ranging from 1 to $2^{10}$ in log-scale. It is easy to see the exponential difference between the Label query complexity in blue, and the Comparison query complexity in orange. Further, Theorem 3.7 also suggests that Perfect-Learning$(N, \text{Comparison}, d)$ should scale near linearly in dimension. The right graph in Figure 1 confirms that this is true in practice as well.

## 6 Further Directions

### 6.1 Average Inference Dimension and Enriched Queries

KLMZ [5] propose looking for a simple set of queries with finite inference dimension $k$ for $d$-dimensional linear separators. In particular, they suggest looking at extending to t-local relative

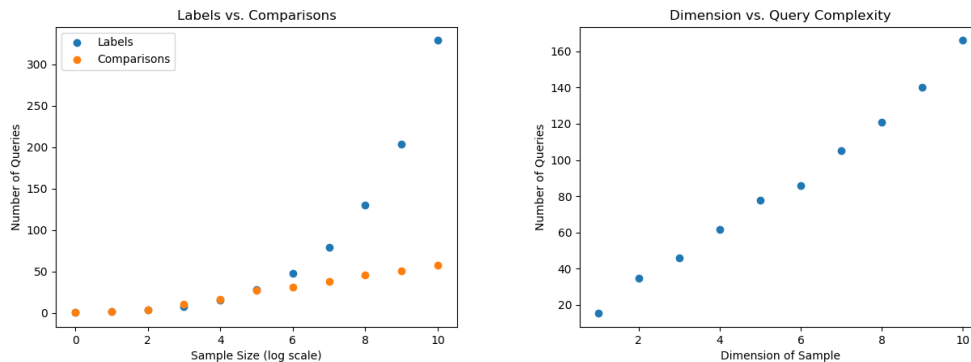

Figure 1: Left: log-scale comparison of Perfect-Learning($N$, Label, 3, 1) and Perfect-Learning($N$, Comparison, 3, 2). Right: scaling of Perfect-Learning(256, Comparison, $d$) with dimension.

queries, questions which ask comparative questions about $t$ points. Unfortunately, simple generalizations of comparison queries seem to fail, but the problem of analyzing their average inference dimension remains open. When moving from 1-local to 2-local queries, our average inference dimension improved from:

$$2^{-\tilde{O}(n)} \to 2^{-\tilde{O}(n^2)}$$

If there exist simple relative t-local queries with average inference dimension $2^{-\tilde{O}(n^t)}$ over some distribution $D$, then it would imply a passive RPU-learning algorithm over $D$ with sample complexity

$$n(\varepsilon, \delta) = O\left(\frac{\log\left(\frac{1}{\varepsilon}\right)^{1/(t-1)}}{\varepsilon} \log\left(\frac{1}{\delta}\right)\right)$$

and query complexity

$$q(\varepsilon, \delta) \leq O\left(2 f_Q\left(4 \log^{1/(t-1)}(n)\right) \log\left(\frac{1}{\delta}\right) \log(n)\right)$$

One such candidate 3-local query given points $x_1, x_2$, and $x_3$ is the question: is $x_1$ closer to $x_2$, or $x_3$? KLMZ suggest looking into this query in particular, and other similar types of relative queries are studied in [29–35].

## 6.2 Noisy and Agnostic Learning

The models we have proposed in this paper are unrealistic in the fact that they assume a perfect oracle. RPU-learning in particular must be noiseless due to its zero-error nature. This raises a natural question: *can inference dimension techniques be applied in a noisy or non-realizable setting?* Hopkins, Kane, Lovett, and Mahajan [13] recently made progress in this direction, introducing a relaxed version of RPU-learning called Almost Reliable and Probably Useful learning. They are able to provide learning algorithms under the popular [7, 2, 28, 36–39] Massart [40] and Tsybakov noise [41, 42] models.

However, many problems in this direction remain completely open, such as agnostic or more adversarial noise. It remains unclear whether inference based techniques are robust to these settings, since small adversarial adjustments to the inference LP can cause substantial corruption to its output.

## Broader Impact

Since this work is theoretical in nature, we do not foresee any particular applications.

## Acknowledgments and Disclosure of Funding

Max Hopkins was supported by NSF Award DGE-1650112. Daniel Kane was supported by NSF Award CCF-1553288 (CAREER) and a Sloan Research Fellowship. Shachar Lovett was supported by NSF Award CCF-1909634.

## Footnotes

[1]Formally, $n(\varepsilon, \delta)$ must also be polynomial in a number of parameters of $C$.

[2]We note that in this version of the model, the learner must know the support of the distribution. Since we only use the model for lower bounds, we lose no generality by making this assumption.

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
