[Supplementary Material · Appendix_Camera_Ready.pdf]

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

These two bounds are a generalization of the technique employed by El-Yaniv and Wiener [4] to prove lower bounds for a specific algorithm, and apply to any learner. We further show that this bound is tight up to a logarithmic factor. For passive RPU-learning with comparison queries, we will simply inherit the lower bound from the PAC model (Proposition 1.3).

**Corollary 1.7.** *For small enough $\varepsilon$, and $\delta = \frac{3}{8}$, any algorithm that Comparison-Passive-RPU learns* $(B^d, \mathbb{R}^d, H_d)$ *must use at least*

$$n(\varepsilon, 3/8) = \Omega \left( \frac{1}{\varepsilon} \right)$$

*samples.*

Note that unlike for label queries, this lower bound is not exponential in dimension. In fact, we will show that this bound is tight up to a linear factor in dimension, and further that employing comparison queries in general shifts the RPU model from being intractable to losing only a logarithmic factor over PAC-learning in both the passive and active regimes. We need one definition: two distributions $D, D'$ over $\mathbb{R}^d$ are affinely equivalent if there is an invertible affine map $f : \mathbb{R}^d \to \mathbb{R}^d$ such that $D(x) = D'(f(x))$.

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

Theorems 1.8 and 1.9 follow from the fact that small average inference dimension implies that finite samples will have low inference dimension with good probability (Observation 4.6). Our main technical contribution lies in proving a structural result (Theorem 4.10): that the average inference dimension of $(D, \mathbb{R}^d, H_d)$ with respect to comparison queries is superexponentially small, $2^{-\Omega_d(n^2)}$, as long as $D$ satisfies the weak distributional requirements outlined in Theorem 1.8.

## 2 Background: Inference Dimension

Before proving our main results, we detail some additional background in inference dimension that is necessary for our RPU-learning techniques. First, we review the inference-dimension based upper and lower bounds of KLMZ [5]. Let $f_Q(k)$ be the number of oracle queries required to answer all queries on a sample of size $k$ in the worst case (e.g. $f_Q(k) = O(k \log(k))$ for comparison queries via sorting). Finite inference dimension implies the following upper bound:

**Theorem 2.1** ([5])**.** *Let $k$ denote the inference dimension of $(X, H)$ with query set $Q$. Then the expected query complexity of $(X, H)$ for $|X| = n$ is:*

$$\mathbb{E}[q(n)] \leq 2f_Q(4k)\log(n).$$

Further, infinite inference dimension provides a lower bound:

**Theorem 2.2** ([5])**.** *Assume that the inference dimension of $(X, H)$ with query set $Q$ is $> k$. Then for $\varepsilon = \frac{1}{k}, \delta = \frac{1}{6}$, the sample complexity of Q-Pool-PAC learning $(X, H)$ is:*

$$q(\varepsilon, 1/6) = \Omega(1/\varepsilon).$$

As the name would suggest, the upper bound derived via inference dimension is based upon a reliable learner that infers a large number of points given a small sample. While not explicitly stated in [5], it follows from the same argument that finite inference dimension gives an upper bound on the sample complexity of RPU-learning:

**Corollary 2.3.** *Let $k$ denote the inference dimension of $(X, H)$ with query set $Q$. Then the sample complexity of Q-Passive-RPU learning $(X, H)$ is:*

$$\mathbb{E}[n(\varepsilon)] = O\left(\frac{k}{\varepsilon}\right).$$

# 3 PAC Learning with Comparison Queries

In this section we study PAC learning with comparison queries in both the passive and active cases.

## 3.1 Lower Bounds

To begin, we prove that over a uniform distribution on a unit ball, learning linear separators with only label queries is hard.

**Proposition 3.1** (Restatement of Proposition 1.2)**.** *For small enough $\varepsilon$, and $\delta = \frac{1}{2}$, the query complexity of Label-MQS-PAC learning $(B^d, \mathbb{R}^d, H_d)$ is:*

$$q(\varepsilon, 1/2) = \Omega_d\left(\left(\frac{1}{\varepsilon}\right)^{\frac{d-1}{d+1}}\right).$$

*Proof.* This follows from a packing argument. It is well known (see e.g. [28]) that for small enough $\varepsilon$, it is possible to pack $\Omega_d\left(\left(\frac{1}{\varepsilon}\right)^{\frac{d-1}{d+1}}\right)$ disjoint spherical caps (intersections of halfspaces with $B^d$) of volume $2\varepsilon$ onto $B^d$. By Yao's Minimax Theorem [29], we may consider a randomized strategy from the adversary such that any deterministic strategy of the learner will fail with constant probability. Consider an adversary which picks one of the disjoint spherical caps to be negative uniformly at random. If the learner queries only $O_d\left(\left(\frac{1}{\varepsilon}\right)^{\frac{d-1}{d+1}}\right)$ points, for a small enough constant and $\varepsilon$, any strategy will uncover the negative cap with at most some constant, say less than 25% probability. Since for small enough $\varepsilon$ there will be at least three remaining caps in which the learner never queried a point, the probability that the learner outputs the correct negative cap (which is necessary to learn up to error $\varepsilon$), is at most $1/3$ due to uniform distribution of the negative cap. Thus alltogether the learner will fail with probability at least $1/2$. □

To show that our exponential improvement comes from the use of comparisons in combination with active learning, we will prove that using comparisons coupled with passive learning provides no improvement.

**Proposition 3.2** (Restatement of Proposition 1.3)**.** *For small enough $\varepsilon$, and $\delta = \frac{3}{8}$, any algorithm that passively learns $(B^d, \mathbb{R}^d, H_d)$ with comparison queries must use at least*

$$n(\varepsilon, \delta) = \Omega\left(\frac{1}{\varepsilon}\right)$$

*samples.*

*Proof.* Let $h_{2\varepsilon}$ be a hyperplane which forms a spherical cap $c$ of measure $2\varepsilon$, and $h$ be the parallel hyperplane tangent to this cap. By Yao's Minimax Theorem [29], we consider an adversary which chooses uniformly between $h_{2\varepsilon}$ and $h$. Given $k$ uniform samples from $B^d$, the probability that at least one point lands inside the cap $c$ is $\leq 2k\varepsilon$. Let

$$k = o\left(\frac{1}{\varepsilon}\right),$$

then for small enough $\varepsilon$, this probability is $\leq 1/4$. Say no sample lands in $c$, then $h_{2\varepsilon}$ and $h$ are completely indistinguishable by label or comparison queries. Any hypothesis chosen by the learner must label at least half of $c$ positive or negative, and will thus have error $\geq \varepsilon$ with either $h_{2\varepsilon}$ or $h$. Since the distribution over these hyperplanes is uniform, the learner fails with probability at least 50%. Thus in total the probability that the learner fails is at least $\frac{3}{4} \cdot \frac{1}{2} = \frac{3}{8}$ □

Together, these lower bounds show it is only the combination of active learning and comparison queries which provides an exponential improvement.

## 3.2 Upper Bounds

For completeness, we will begin by showing that Proposition 1.2 is tight for $d = 2$ before moving to our main result for the section.

**Proposition 3.3.** *The query complexity of Label-MQS-PAC learning $(B^2, \mathbb{R}^2, H_2)$ is:*

$$q(\varepsilon, 0) = O\left(\left(\frac{1}{\varepsilon}\right)^{\frac{1}{3}}\right).$$

*Proof.* To begin, we will show that selecting $k = O\left(\left(\frac{1}{\varepsilon}\right)^{\frac{1}{3}}\right)$ points along the boundary of $B^2$ in a regular fashion (such that their convex hull is the regular $k$ sided polygon) is enough if all such points have the same label. This follows from the fact that each cap created by the polygon has area and thus probability mass

$$Area(Cap) = \frac{1}{2}(2\pi/k - \sin(2\pi/k)).$$

Taylor approximating sine shows that picking $k = O\left(\left(\frac{1}{\varepsilon}\right)^{\frac{1}{3}}\right)$ gives Area(Cap) $< \varepsilon$. If all $k$ points are of the same sign (say 1), a hyperplane can only cut through one such cap, and thus labeling the entire disk 1.
Thus we have reduced to the case where there are one or more points of differing signs. In this scenario, there will be exactly two edges where connected vertices are of different signs, which denotes that the hyperplane passes through both edges. Next, on each of the two caps associated with these edges, we query $O(\log(1/\varepsilon))$ points in order to find the crossing point of the hyperplane via binary search up to an accuracy of $\varepsilon/2$. This reduces the area of unknown labels to the strip connecting these two $< \varepsilon/2$ arcs, which has $< \varepsilon$ probability mass. Picking any consistent hyperplane then finishes the proof. □

Now we will show that active learning with comparison queries in the PAC-learning model exponentially improves over the passive and label regimes. Our work is closely related to the algorithm of Balcan and Long [2], and relies on using comparison queries to reduce to a combination of their algorithm and thresholding. Our bounds will relate to a general set of distributions called isotropic (0-centered, identity variance) log-concave distributions, distributions whose density function $f$ may be written as $e^{g(x)}$ for some concave function $g$. log-concavity generalizes many natural distributions such as gaussians and convex sets. To begin, we will need a few statements regarding isotropic log-concave distributions proved initially by Lovasz and Vempala [30], and Klivans, Long, and Tang [31] (here we include additional facts we require for RPU-learning later on).

**Fact 3.4** ([30, 31]). *Let $D$ be an arbitrary log-concave distribution in $\mathbb{R}^d$ with probability density function $f$, and $u, v$ normal vectors of homogeneous hyperplanes. The following statements hold where 3,4,5, and 6 assume $D$ is isotropic:*

---

**Algorithm 1:** Comparison-Pool-PAC learn $(D, \mathbb{R}^d, H_d)$

---

**1** $N = O\left(\frac{1}{\varepsilon}\right)$; shift_list = [];

**2** normal_vector = B-L$\left(Iso(D-D), O\left(\frac{\log(1/\varepsilon)}{\varepsilon}\right), \delta\right)$;

**3 for** *i in range* $O(\log(1/\delta))$ **do**

**4**     $S \sim D^N$;

**5**     $S = \text{Project}(S, \text{normal\_vector})$;

**6**     shift_list.add(Threshold(S));

**7 end**

**8** Return $h = \langle \text{normal\_vector}, \cdot \rangle + \text{median(shift\_list)}$

---

Figure 1: Algorithm for Comparison-Pool-PAC learning hyperplanes over a log-concave distribution $D$. Our algorithm references three sub-routines. The first, B-L$(D, \varepsilon, \delta)$, is the algorithm from Theorem 3.5. The second is Project(sample,vector), which simply projects each point in a sample onto the given vector. The third is Threshold($S$), which produces a threshold consistent with labeling $S$ by binary search.

1. $D - D$, the difference of i.i.d pairs, is log-concave

2. $D$ may be affinely transformed to an isotropic distribution $Iso(D)$

3. There exists a universal constant $c$ s.t. the angle between any $u$ and $v$, denoted $\theta(u, v)$, satisfies $c\theta(u, v) \leq \Pr_{x \sim D}[sgn(\langle x, v \rangle) \neq sgn(\langle x, u \rangle)]$

4. $\forall a > 0, \Pr_{x \in D}[\|x\| \geq a] \leq e^{-\frac{a}{\sqrt{d}}+1}$

5. All marginals of $D$ are isotropic log-concave

6. If $d = 1, \Pr_{x \in D}[x \in [a, b]] \leq |b - a|$

We will additionally need Balcan and Long's [2] query optimal algorithm for label-Pool-PAC learning homogeneous hyperplanes[3].

**Theorem 3.5** (Theorem 5 [2])**.** *Let $D$ be a log-concave distribution over $\mathbb{R}^d$. The query complexity of Label-Pool-PAC learning $(D, \mathbb{R}^d, H_d^0)$ is*

$$q(\varepsilon, \delta) = O\left(\left(\left(d + \log\left(\frac{1}{\delta}\right)\right) + \log\log\left(\frac{1}{\varepsilon}\right)\right)\log\left(\frac{1}{\varepsilon}\right)\right),$$

*where $H_d^0$ is the class of homogeneous hyperplanes.*

Using these facts, we will give an upper bound for the Pool-based model assuming a pool of $\text{Poly}(1/\varepsilon, \log(1/\delta))$ unlabeled samples. For a sketch of the algorithm, see Figure 1.

**Theorem 3.6** (Restatement of Theorem 1.4)**.** *Let $D$ be a log-concave distribution over $\mathbb{R}^d$. The query complexity of Comparison-Pool-PAC learning $(D, \mathbb{R}^d, H_d)$ is*

$$q(\varepsilon, \delta) = O\left(\left(\left(d + \log\left(\frac{1}{\delta}\right)\right) + \log\log\left(\frac{1}{\varepsilon}\right)\right)\log\left(\frac{1}{\varepsilon}\right)\right).$$

*Proof.* Recall that $D$ may be affinely transformed into an isotropic distribution $Iso(D)$. Further, we may simulate queries over $Iso(D)$ by applying the same transformation to our samples, and after learning over $Iso(D)$, we may transform our learner back to $D$. Thus learning $Iso(D)$ is equivalent to learning $D$, and we will assume $D$ is isotropic without loss of generality. Our algorithm will first learn a "homogenized" version of the hidden separator $h = \langle v, \cdot \rangle + b$ via Balcan and Long's algorithm, thereby reducing to thresholding.

Note that comparison queries on the difference of points $x, y \in D$ is equivalent to a label query on the point $x - y$ on the *homogeneous* hyperplane with normal vector $v$:

$$h(x) - h(y) = (\langle v, x \rangle + b) - (\langle v, y \rangle + b) = \langle v, x - y \rangle.$$

We begin by drawing samples from the log-concave distribution $D - D$ and then apply Balcan and Long's algorithm [2] to learn the homogenized version of $h$ ($\langle v, \cdot \rangle$) up to $O\left(\frac{\varepsilon}{\log(1/\varepsilon)}\right)$ error with probability $1 - \delta$ using only

$$O\left(\left(d + \log\left(\frac{1}{\delta}\right) + \log\log\left(\frac{1}{\varepsilon}\right)\right)\log\left(\frac{1}{\varepsilon}\right)\right)$$

comparison queries. Further, since the constant $c$ given in item 2 of Fact 3.4 is universal, this means any separator output by the algorithm has a normal vector $u$ with angle

$$\theta(u, v) = O\left(\frac{\varepsilon}{\log(1/\varepsilon)}\right).$$

Having learned an approximation to $v$, we turn our attention to approximating $b$. Consider the set of points on which $u$ and $v$ disagree, that is:

$$Dis = \{x : sgn(\langle v, x \rangle + b) \neq sgn(\langle u, x \rangle + b)\}$$

To find an approximation for $b$, we need to show that there will be correctly labeled points close to the threshold. To this end, let $\alpha = \varepsilon/8$ and define $b_{\pm\alpha}$ such that:

$$D(\{y : \alpha < \langle u, y \rangle + b < b_\alpha\}) = \alpha$$
$$D(\{y : b_{-\alpha} < \langle u, y \rangle + b < -\alpha\}) = \alpha$$

We will show that drawing a sample $S$ of $O\left(\frac{1}{\varepsilon}\right)$ points, the following three statements hold with at least $2/3$ probability:

1. $\exists x_1 \in S : \alpha < \langle u, x_1 \rangle + b < b_\alpha$

2. $\exists x_2 \in S : b_{-\alpha} < \langle u, x_2 \rangle + b < -\alpha$

3. $\forall x \in Dis \cap S, |\langle u, x \rangle + b| < \alpha$

Since the measure of the regions defined in statements 1 and 2 is $\varepsilon/4$, the probability that $S$ does not have at least one point in both regions is $\leq 2 * (1 - \alpha)^{|S|} \leq 1/6$ with an appropriate constant.

To prove the third statement, assume for contradiction that there exists $x \in Dis \cap S$ such that $|\langle u, x \rangle + b| > \varepsilon/4$. Because $\langle u, x \rangle + b$ and $\langle v, x \rangle + b$ differ in sign, this implies that $|\langle u - v, x \rangle| = |\langle u - v, x_{u,v} \rangle| > \alpha$, where $x_{u,v}$ is the projection of $x$ onto the plane spanned by u and v. We can bound the probability of this event occurring by the concentration of isotropic log-concave distributions:

$$Pr[|\langle u - v, x_{u,v} \rangle| > \alpha] \leq e^{-\Omega\left(\frac{\varepsilon}{|u-v|}\right)}. \tag{1}$$

Because we have bounded the angle between $u$ and $v$, with a large enough constant for $\theta$ we have:

$$|u - v| \leq O\left(\frac{\varepsilon}{\log(|S|)}\right).$$

Then with a large enough constant for $\theta$, union bounding over $Dis \cap S$ gives that the third statement occurs with probability at most $1/6$.

We have proved that with probability $2/3$, statements 1,2, and 3 hold. Further, if these statements hold, any hyperplane $\langle u, \cdot \rangle + b'$ we pick consistent with thresholding will disagree on at most $\varepsilon/4$ probability mass from $\langle u, \cdot \rangle + b$ due to the anti-concentration of isotropic log-concave distributions and the definition of $b_{\pm\alpha}$.

Further, repeating this process $O(\log(1/\delta))$ times and taking the median shift value $b'$ gives the same statement with probability at least $1 - \delta$ by a Chernoff bound. Note that the number of queries made in this step is dominated by the number of queries to learn $u$.

Finally, we need to analyze the error of our proposed hyperplane $\langle u, \cdot \rangle + b'$. We have already proved that the error between this and $\langle u, \cdot \rangle + b$ is $\leq \varepsilon/4$ with probability at least $1 - \delta$, so it is enough to show that $D(Dis) \leq 3\varepsilon/4$. This follows similarly to statement 3 above. The portion of Dis satisfying $|\langle u, x \rangle + b| \leq \alpha$ has probability mass at most $\varepsilon/4$ by anti-concentration. With a large enough constant for $\theta$, the remainder of Dis has mass at most $\varepsilon/2$ by (1). Then in total, with probability $1 - 2\delta$, $\langle u, \cdot \rangle + b'$ has error at most $\varepsilon$. $\qquad \square$

Balcan and Long [2] provide a lower bound on query complexity for log-concave distributions and oracles for any binary query of $\Omega(d \log(\frac{1}{\varepsilon}))$, so this algorithm is tight up to logarithmic factors.

# 4  RPU Learning with Comparison Queries

Kivinen [12] showed that RPU-learning is intractable for nice concept classes even under simple distributions when restricted to label queries. We will confirm that RPU-learning linear separators with only label queries is intractable in high dimensions, but can be made efficient in both the passive and active regimes via comparison queries.

## 4.1  Lower bounds

In the passive, label-only case, RPU-learning is lower bounded by the expected number of vertices on a random polytope drawn from our distribution $D$. For simple distributions such as uniform over the unit ball, this gives sample complexity which is exponential in dimension, making RPU-learning impractical for any sort of high-dimensional data.

**Definition 4.1.** *Given a distribution $D$ and parameter $\varepsilon > 0$, we denote by $v_D(\varepsilon)$ the minimum size of a sample $S$ drawn i.i.d from $D$ such that the expected measure of the convex hull of $S$, which we denote $E(D, n)$ for $|S| = n$, is $\geq 1 - \varepsilon$.*

The quantity $v_D(\varepsilon)$, which has been studied in computational geometry for decades [33, 34], lower bounds Label-Passive-RPU Learning, and in some cases provides a matching upper bound up to log factors.

**Proposition 4.2.** *Let $D$ be any distribution on $\mathbb{R}^d$. The expected sample complexity of Label-Passive-RPU-learning $(D, \mathbb{R}^d, H_d)$ is:*

$$n(\varepsilon, 1/3) = \Omega \left( \frac{v_D(2\varepsilon)}{\log(1/\varepsilon)} \right).$$

*Proof.* The proof follows from considering a setting in which the learner will almost always draw positive points, and therefore cannot infer anything outside of their convex hull. More formally, for any $\delta > 0$ and sample size $n$, there exists some radius $r_{\delta,n}$ such that the probability that a sample $S \sim D^n$ contains any point outside the ball of radius $r_{\delta,n}$, $B^d(r_{\delta,n})$, is less than $\delta$. By Yao's Minimax Theorem [29], it is sufficient to consider an adversary who picks some hyperplane tangent to $B^d(r_{\delta,n})$ with probability $1 - \delta$ (labeling it entirely positive), and otherwise chooses a hyperplane uniformly from $S^d \times [-r_{\delta,n}, r_{\delta,n}]$. Notice that if the adversary chooses the tangent hyperplane and the learner draws a sample $S$ entirely within the ball, for any point $x$ outside the convex hull of $S$ there exist hyperplanes within the support of the adversary's distribution that are consistent on $S$ but differ on $x$.

Recall that $v_D(\varepsilon)$ is the minimum size of the sample $S$ which needs to be drawn such that $1 - E(D, n)$ is $\leq \varepsilon$ in expectation. Consider drawing a sample $S$ of size $n = v_D(2\varepsilon) - 1$. The expected measure $E(D, n)$ is then

$$E(D, n) < 1 - 2\varepsilon.$$

This in turn implies a bound by the Markov inequality on the probability of the measure of the convex hull of a given sample, which we denote $V(S)$:

$$Pr_{S \sim D^n}[V(S) \geq 1 - \varepsilon] \leq \frac{1 - 2\varepsilon}{1 - \varepsilon} = 1 - \frac{\varepsilon}{1 - \varepsilon}.$$

Now consider the following relation between samples of size $n$ and $\lfloor n/k \rfloor$, which follows by viewing our size $n$ sample as $k$ distinct samples of size at least $n/k$:

$$1 - Pr_{S \sim D^{\lfloor n/k \rfloor}}[V(S) < 1 - \varepsilon]^k \leq Pr_{S \sim D^n}[V(S) \geq 1 - \varepsilon].$$

Combining these results and letting $k = \log(1/\varepsilon)$:

$$Pr_{S \sim D^{\lfloor n/k \rfloor}}[V(S) < 1 - \varepsilon] \geq \left(\frac{\varepsilon}{1 - \varepsilon}\right)^{1/k} \geq 1/2.$$

To force any learner to fail on a sample, we need two conditions: first that the measure of the convex hull is $< 1 - \varepsilon$, and second that all points lie in $B^d(r_{\delta,n})$. Since the latter occurs with probability $1 - 2\delta$, picking $\delta < 1/12$ then gives the desired success bound:

$$Pr_{S \sim D^{\lfloor n/\log(1/\varepsilon) \rfloor}}[(V(S) \geq 1 - \varepsilon) \vee (\exists x \in S : x \notin B^d(r_{1/12,n})] \leq 1/2 + 2\delta < 2/3.$$

$\square$

Further, for simple distributions such as uniform over a ball, this bound is tight up to a $\log^2$ factor.

**Proposition 4.3.** *The sample complexity of Label-Passive-RPU learning $(B^d, \mathbb{R}^d, H_d)$ is:*

$$\mathbb{E}[n(\varepsilon)] = O\left(\log(d/\varepsilon)v_{B^d}\left(\frac{\varepsilon}{2}\right)\right) = O_d\left(\log(1/\varepsilon)\left(\frac{1}{\varepsilon}\right)^{\frac{d+1}{2}}\right).$$

*Proof.* We will begin by computing $v_{B^d}(\varepsilon)$ for a ball. The expected measure of a sample drawn randomly from $B^d$ is computed in [35], and given by

$$E(B^d, n) = 1 - c(d)n^{-\frac{2}{d+1}},$$

where $c(d)$ is a constant depending only on dimension. Setting $c(d)n^{\frac{-2}{d+1}} = \varepsilon$ then gives:

$$v_{B^d}(\varepsilon) = \left(\left(\frac{c(d)}{\varepsilon}\right)^{\frac{d+1}{2}}\right)$$

Given a sample $S$ of size $O(\log(1/\delta)n)$, let $S_p$ denote the subset of positively labeled points, and $S_n$ negatively labeled. We can infer at least the points inside the convex hulls of $S_p$ and $S_n$. Our goal is to show that, with high probability, the measure of $M = \text{ConvHull}(S_p) \cup \text{ConvHull}(S_n)$ is $\geq 1 - \varepsilon$. To show this, we will employ the fact [33] that the expected measure of the convex hull of a sample of size $n$ uniformly drawn from any convex body $K$ is lower-bounded by:

$$E(K, n) = 1 - c(d)n^{-\frac{2}{d+1}}.$$

Given this, let $P$ of measure $p$ be the set of positive points, and $N$ the negative points with measure $1 - p$. Since we have drawn $O(\log(1/\delta)n)$ points, with probability $\geq 1 - \delta$ we will have at least $pn$ points from $P$, and at least $(1 - p)n$ points from $N$. Given this many points, the expected value of our inferred mass $M$ is:

$$\mathbb{E}[M] \geq pE(P, pn) + (1 - p)E(N, (1 - p)n)$$
$$= 1 - c(d)\left(p(pn)^{-2/(d+1)} + (1 - p)((1 - p)n)^{-2/(d+1)}\right).$$

This function is minimized at $p = .5$, and plugging in $p = .5$, $n = 2v_{B^d}\left(\frac{\varepsilon}{2}\right)$ gives $\mathbb{E}[M] \geq 1 - \frac{2d-1}{2d}\varepsilon$.

However, since we have conditioned on enough points being drawn from P and N, we are not done. This occurs across at least a $1 - \delta$ percent of our samples, meaning that if we assume the inferred mass $M$ is 0 on other samples, our expected error (for a large enough constant on our number of samples) will be at most:

$$1 - \mathbb{E}[M] = (1 - \delta)\frac{(2d - 1)\varepsilon}{2d} + \delta.$$

Setting $\delta = \varepsilon/(2d)$ is enough to drop the error below $\varepsilon$, and gives the number of samples as

$$O\left(\log(d/\varepsilon)v_{B^d}\left(\frac{\varepsilon}{2}\right)\right).$$

$\square$

In the active regime, this sort of bound is complicated by the fact that we are less interested in the number of points drawn than labeled. If we were restricted to only drawing $\mathbb{E}[n(\varepsilon)]$ points, we could repeat the same argument in combination with the expected number of vertices to get a bound. However, with a larger pool of allowed points, the pertinent question becomes the maximum rather than expected measure of the convex hull. In cases such as the unit ball, these actually give about the same result.

**Proposition 4.4** (Restatement of Proposition 1.6). *For all $\delta < 1$, the query complexity of Label-MQS-RPU learning $(B^d, \mathbb{R}^d, H_d)$ is:*

$$q(\varepsilon, \delta) = \Omega_d\left(\left(\frac{1}{\varepsilon}\right)^{\frac{d-1}{2}}\right)$$

*Proof.* The maximum volume of the convex hull of $n$ points in $B^d$ is [34]

$$\max_{S, |S|=n}(\text{Vol}(\text{ConvHull}(S))) = 1 - \theta_d\left(n^{-\frac{2}{d-1}}\right).$$

Notice here the difference from the random case in the exponent, which comes from the fact that we are only counting the expected $\theta_d\left(n^{\frac{d-1}{d+1}}\right)$ vertices on the boundary of the hull of the sample. The lower bound is then implied by the same adversary strategy as in Proposition 4.2, since for small enough $\varepsilon$, the convex hull of any set of $o_d\left(\left(\frac{1}{\varepsilon}\right)^{\frac{d-1}{2}}\right)$ points has less than $1 - \varepsilon$ probability mass. $\square$

## 4.2 Upper bounds

Our positive results for comparison based RPU-learning rely on weakening the concept of inference dimension to be distribution dependent. With this in mind, we introduce average inference dimension:

**Definition 4.5** (Average Inference Dimension). *We say $(D, X, H)$ has average inference dimension $g(n)$, if:*

$$\forall h \in H, Pr_{S \sim D^n}[\nexists x \text{ s.t. } S - \{x\} \to_h x] \leq g(n).$$

In other words, the probability that we cannot infer a point from a randomly drawn sample of size n is bounded by its average inference dimension $g(n)$. There is a simple average-case to worst-case reduction for average inference dimension via a union bound:

**Observation 4.6.** *Let $(D, X, H)$ have average inference dimension $g(n)$, and $S \sim D^n$. Then $(S, H)$ has inference dimension $k$ with probability:*

$$Pr[\text{Inference dimension of } (S, H) \leq k] \geq 1 - \binom{n}{k}g(k).$$

*Proof.* The probability that a fixed subset $S' \subset S$ of size $k$ does not have a point $x$ s.t. $S - \{x\} \to_h x$ is at most $g(k)$. Union bounding over all $\binom{n}{k}$ subsets gives the desired result. $\square$

This reduction allows us to apply inference dimension in both the active and passive distributional cases. This is due in part to the fact that the boosting algorithm proposed by KLMZ [5][Theorem 3.2] is reliable even when given the wrong inference dimension as input–their core algorithm simply runs a linear program whose constraints are given by query responses and thus never errs. As a result, we may plug this reduction directly into their algorithm.

**Corollary 4.7.** *Given a query set $Q$, let $f_Q(n)$ be the number of queries required to answer all questions on a sample of size $n$. Let $(D, X, H)$ have average inference dimension $g(n)$, then there exists an RPU-learner $A$ with coverage*

$$\mathbb{E}[C(A)] = \max_{k \leq n} \left( 1 - \binom{n}{k} g(k) \right) \frac{n-k}{n}$$

*after drawing $n$ points. Further, the expected query complexity of actively RPU-learning a finite sample $S \sim D^n$ is*

$$\mathbb{E}[q(n)] \leq \min_{k \leq n} 2 f_Q(4k) \log(n) \left( 1 - g(k) \binom{n}{k} \right) + n g(k) \binom{n}{k}$$

*Proof.* For the first fact, we will appeal to the symmetry argument of [5]. Consider a reliable learner $A$ which takes in a sample $S$ of size $n - 1$ and infers all possible points in $D$. To compute coverage, we want to know the probability a random point $x \sim D$ is inferred by $A$. Since $S$ was randomly drawn from $D$, this is the same as computing the probability that any point in $S \cup \{x\}$ can be inferred from $S$. By Observation 4.6, the probability that $S \cup \{x\}$ has inference dimension $k$ is

$$\left( 1 - \binom{n}{k} g(k) \right).$$

Since $x$ could equally well have been any point in $S$ by symmetry, if $S$ has inference dimension $k$ the coverage will be at least $\frac{n-k}{n}$ [5]. Since this occurs with probability at least $1 - \binom{n}{k} g(k)$ by Observation 4.6, the expected coverage of $A$ is at least

$$\mathbb{E}[C(A)] \geq \left( 1 - \binom{n}{k} g(k) \right) \frac{n-k}{n}.$$

The second statement follows from a similar argument. If $S$ has inference dimension $k$, then by Theorem 2.1 the expected query complexity is at most $2 f_Q(4k) \log(n)$. For a given $k$, the expected query complexity is then bounded by:

$$\mathbb{E}[q(n)] \leq 2 f_Q(4k) \log(n) \Pr[\text{S has inference dimension} \leq k] + n \Pr[\text{S has inference dimension} > k].$$

Plugging in Observation 4.6 and minimizing over $k$ then gives the desired result. $\qquad\square$

In fact, this lemma shows that RPU-learning $(D, X, H)$ with inverse super-exponential average inference dimension loses only log factors over passive or active PAC-learning. Asking for such small average inference dimension may seem unreasonable, but something as simple as label queries on a uniform distributions over convex sets has average inference dimension $2^{-\Theta(n \log(n))}$ with respect to linear separators [36].

**Corollary 4.8.** *Given a query set $Q$, let $f_Q(n)$ be the number of queries required to answer all questions on a sample of size $n$. For any $\alpha > 0$, let $(D, X, H)$ have average inference dimension $g(n) \leq 2^{-\Omega(n^{1+\alpha})}$. Then the expected sample complexity of Q-Pool-RPU learning is:*

$$\mathbb{E}[n(\varepsilon)] = O\left( \frac{\log(\frac{1}{\varepsilon})^{1/\alpha}}{\varepsilon} \right).$$

*Further, the expected query complexity of actively learning a finite sample $S \sim D^n$ is:*

$$\mathbb{E}[q(n)] \leq 2 f_Q \left( O\left( \log^{1/\alpha}(n) \right) \right) \log(n).$$

*Proof.* Both results follow from the fact that setting the average inference dimension $k$ to $O\left(\log(n)^{1/\alpha}\right)$ gives

$$\left(1 - \binom{n}{k} g(k)\right) = 1 - O\left(\frac{1}{n}\right).$$

Then for the sample complexity, it is enough to plug this into Corollary 4.7 and let $n$ be

$$n = O\left(\frac{\log(\frac{1}{\varepsilon})^{1/\alpha}}{\varepsilon}\right).$$

Plugging this into the query complexity sets the latter term from Corollary 4.7 to 1, giving:

$$\mathbb{E}[q(n)] \leq 2f_Q\left(O\left(\log^{1/\alpha}(n)\right)\right)\log(n).$$

$\square$

We will show that by employing comparison queries we can improve the average inference dimension of linear separators from $2^{\Omega(-n\log(n))}$ to $2^{-\Omega(n^2)}$, but first we will need to review a result on inference dimension from [5].

**Theorem 4.9** (Theorem 4.7 [5]). *Given a set $X \subseteq \mathbb{R}^d$, we define the minimal-ratio of $X$ with respect to a hyperplane $h \in H_d$ as:*

$$\frac{\min_{x \in X} |h(x)|}{\max_{x \in X} |h(x)|}.$$

*In other words, the minimal-ratio is a normalized version of margin, a common tool in learning algorithms. Given $X$, define $H_{d,\eta} \subseteq H_d$ to be the subset of hyperplanes with minimal ratio $\eta$ with respect to $X$. The inference dimension of $(X, H_{d,\eta})$ is then:*

$$k \leq 10d\log(d+1)\log(2\eta^{-1}).$$

Our strategy to prove the average inference dimension of comparison queries follows via a reduction to minimal-ratio. Informally, our strategy is very simple. We will argue that, with high probability, throwing out the closest and furthest points from any classifier leaves a set with large minimal-ratio. We will show this in three main steps.

**Step 1:** Assuming concentration of our distribution, a large number of points are contained inside a ball. We will use this to bound the maximum function value for a given hyperplane when its furthest points are removed.

**Step 2:** Assuming anti-concentration of our distribution, we will union bound over all hyperplanes to show that they have good margin. In order to do this, we will define the notion of a $\gamma$-strip about a hyperplane h, which is simply h "fattened" by $\gamma$ in both directions. If not too many points lie inside each hyperplane's $\gamma$-strip, then we can be assured when we remove the closest points the remaining set will have margin $\gamma$. Since we cannot union bound over the infinite set of $\gamma$-strips, we will build a $\gamma$-net of the objects and use this instead.

**Step 3:** Combining the above results carefully shows that for any hyperplane, removing the furthest and closest points leaves a subsample of good minimal-ratio. In particular, by making sure the number of remaining points matches the bound on inference dimension given in Theorem 4.9, we can be assured that one of these points may be inferred from the rest as long as our high probability conditions hold.

**Theorem 4.10.** *Let $D$ be a distribution over $\mathbb{R}^d$ affinely equivalent to another with the following properties:*

*1. $\forall \alpha > 0$, $Pr_{x \sim D}[\|x\| > d\alpha] \leq \frac{c_1}{\alpha}$*

*2. $\forall \alpha > 0$, $\langle v, \cdot \rangle + b \in H_d$, $Pr_{x \sim D}[|\langle x, v \rangle + b| \leq \alpha] \leq c_2\alpha$*

*Then for $n = \Omega(d \log^2(d))$, the average inference dimension $g(n)$ of $(D, \mathbb{R}^d, H_d)$ is*

$$g(n) \leq 2^{-\Omega\left(\frac{n^2}{d \log(d)}\right)},$$

*where the constant has logarithmic dependence on $c_1, c_2$.*

*Proof.* To begin, note that since inference is invariant to affine transformation we can assume that our distribution $D$ satisfies properties 1 and 2 without loss of generality. Our argument will hinge on the minimal ratio based inference dimension bound of [5]. Let $k$ denote inference dimension of $(X, H_{d,\eta})$. We begin by drawing a sample $S$ of size $n$, and set our goal minimal-ratio $\eta$ such that $k = n/3$. In particular, it is sufficient to let

$$\eta = 2^{-\theta\left(\frac{n}{d \log(d)}\right)}.$$

We will now prove that for all hyperplanes, removing the closest and furthest $k$ points from $S$ leaves the remaining points with minimal-ratio $\eta$ with high probability.

To begin, we will show that with high probability, $n - k$ points lie inside the ball $B$ of radius $r = 2^{\theta\left(\frac{n}{d \log(d)}\right)}$ about the origin. By condition 1 on our distribution $D$, we know that the probability any $k = n/3$ size subset lies outside radius $r$ is $\leq \left(\frac{c_1 d}{r}\right)^k$. Union bounding over all possible size $k$ subsets then gives:

$$Pr[\exists S' \subseteq S, |S'| = n/3 : \forall x \in S', ||x|| \geq r] \leq \binom{n}{n/3} 2^{-\Omega\left(\frac{n^2}{d \log(d)}\right) + O(n \log(d c_1))}$$

$$\leq 2^{-\Omega\left(\frac{n^2}{d \log(d)}\right)},$$

where the last step follows with $n = \Omega(d \log^2(d))$ and a large enough constant. Assume then that no such subset exists. What implication does this have for the distance of the $k$ furthest points from any given hyperplane? For a given hyperplane $h$, denote the shortest distance between $h$ and any point in $B$ to be $L$. By removing the furthest $k$ points from $h$, we are guaranteed that the maximum distance is $2r + L$. We will separate our analysis into two cases: $L \leq r$ and $L > r$.

In the case that $L \leq r$, our problem reduces to classifiers which intersect the ball $B_2$ of radius $2r$. This further allows us to reduce our question from one of minimal-ratio to margin, as the minimal-ratio is bounded by:

$$\eta \geq \gamma/(4r).$$

Then with the correct parameter setting, it is enough to show that $\gamma \leq r^{-2}$ with high probability for all hyperplanes with $L \leq r$. We will inflate our margin to $\gamma$ by removing the $n/3$ points closest to $h$. It is enough to show that $\forall h$ no subset of $n/3$ points lies in $\{x : h(x) \in [-\gamma, \gamma]\}$, which we will call the $\gamma$-strip, or strip of height $\gamma$, about $h$. Condition 2 gives a bound on this occurring for a given subset of $k$ points and hyperplane $h$, but in this case we must union bound over both subsets and hyperplanes.

Naively, this is a problem, since the set of possible hyperplanes is infinite. However, as we have reduced to hyperplanes intersecting the ball, each is defined by a unit vector $v \in S^d$ and a shift $b \in [-2r, 2r] = [-\gamma^{-1/2}, \gamma^{-1/2}]$. Our strategy will be to build a finite $\gamma$-net $N$ over these strips and show that each point in the net has $O(\gamma^{1/2})$ measure.

Consider the space of normal vectors to our strips, which for now we assume are homogeneous. This is a $d$-unit sphere, which can be covered by at worst $(3\gamma^{-1})^d$ $\gamma$-balls. We can extend this $\gamma$-cover to non-homogeneous strips by placing $4\gamma^{-3/2}$ of these covers at regular intervals along the segment $[-2r, 2r]$. Formally, each point in this cover $N$ corresponds to some hyperplane $h = \langle v, \cdot \rangle + b$, and is comprised of the union $\gamma$-strips nearby $h$:

$$N_{v,b} = \bigcup_{\substack{||v - v'|| \leq \gamma \\ |b - b'| \leq \gamma}} \gamma - \text{strip about } \langle v', \cdot \rangle + b'.$$

What is the measure of $N_{v,b}$? Note that

Figure 2: The above image is the $\gamma$-ball $N_{(1,0)}$ in the simplified homogeneous case. The black strip corresponds to the central hyperplane, and the blue areas denote the strips with close normal vectors. The red dotted line denotes the larger strip in which the $\gamma$-ball lies.

$$N_{v,b} = (N_{v,b} \cap B_2) \cup (N_{v,b} \cap (\mathbb{R}^d \setminus B_2)).$$

We can immediately bound the measure of latter portion by $\frac{c_1 d \sqrt{\gamma}}{2}$ due to concentration. For the former, we will show that $N_{v,b} \cap B_2$ is contained in a small strip with measure bounded by anti-concentration. For a visualization of this, see Figure 2. Since the height of a strip is invariant upon translation, we will let $b = 0$ for simplicity. Consider any $x'$ in the $\gamma$-strip about some hyperplane $\langle v', \cdot \rangle + b' \in N_{v,0}$. Since $v$ is the center of our ball, by definition we have $||v - v'|| \le \gamma$, and $|b'| \le \gamma$. Then for $x'$ in strip $v'$, we can bound $\langle v, x' \rangle$:

$$
\begin{aligned}
|\langle v, x' \rangle| &= |\langle v', x' \rangle + \langle v - v', x' \rangle| \\
&\le 2\gamma + |\langle v - v', x' \rangle| \\
&\le 2\gamma + ||v - v'|| \cdot ||x'|| \\
&\le 2\gamma + \gamma \cdot 2r \\
&= 2(\gamma + \sqrt{\gamma})
\end{aligned}
$$

In other words, this neighborhood of strips lies entirely within the strip about $v$ of height $2(\gamma + \sqrt{\gamma})$, which in turn by condition 2 has measure at most $2c_2(\gamma + \sqrt{\gamma})$.

Finally, note that if no subset of $n/3$ points lies in any $N_{v,b}$, then certainly no such subset lies in a single strip, as $N$ covers all strips. Now we can union bound over subsets and $N$:

$$Pr[\exists N_{v,b} \in N, S' \subseteq S, |S'| = n/3 : \forall x \in S', x \in N_{v,b}] \le \binom{n}{n/3} \left(4(c_1 + c_2)d\gamma^{1/2}\right)^{n/3} (4\gamma^{-1})^{d+5/2}.$$

Recall that $\gamma = 2^{-\theta\left(\frac{n}{d \log(d)}\right)}$. The only term contributing an $n^2$ to the exponent is $\gamma^{n/6}$, and thus plugging in $\gamma$ gives:

$$Pr[\exists N_{v,b} \in N, S' \subseteq S, |S'| = n/3 : \forall x \in S', x \in N_{v,b}] \le 2^{-\Omega\left(\frac{n^2}{d \log(d)}\right)}.$$

The argument for $L > r$ is much simpler. By assuming at least $n - k$ points lie in B, removing the closet k points gives a margin of at least L, and removing the furthest a maximum value of at most $2r + L$. Because

$L > r$, the minimal ratio is bounded by:

$$\eta \geq \frac{r + L}{2r + L} \geq 1/3.$$

Then in total, assuming $|S \cap B| \geq 2n/3$, the probability over samples $S$ that the subsample $S'$ created from removing the closest and furthest $k$ points has minimal-ratio less than $\eta$ is:

$$Pr[\exists h \in H_d : \text{ minimal-ratio of } S' < \eta] \leq 2^{-\Omega\left(\frac{n^2}{d \log(d)}\right)}.$$

Since the probability that $|S \cap B| \geq 2n/3$ is at least $1 - 2^{-\Omega\left(\frac{n^2}{d \log(d)}\right)}$, the above bound holds with no assumption on $|S \cap B|$ as well.

Combining this result together with Theorem 4.9 completes the proof. Let $S'_h$ be the remaining $n/3$ points when the furthest and closest $n/3$ are removed, and assume $S'_h$ has minimal ratio $\eta$. $S'_h$ may thus be viewed as a sample of size $n/3$ from $(S'_h, H_{d,\eta})$. Since $(S'_h, H_{d,\eta})$ has inference dimension $n/3$ for our choice of $\eta$ by Theorem 4.9, $\forall h$ there must exist $x$ s.t. $Q(S'_h - \{x\})$ infers $x$. Thus the probability that we cannot infer a point is upper bounded by $2^{-\Omega\left(\frac{n^2}{d \log(d)}\right)}$ □

Plugging this result into Corollary 4.7 gives our desired guarantee on Comparison-Pool-RPU learning query complexity.

**Theorem 4.11** (Restatement of Theorem 1.8). *Let $D$ be a distribution on $\mathbb{R}^d$ which satisfies the conditions of Theorem 4.10. Then the sample complexity of Comparison-Passive-RPU learning $(D, \mathbb{R}^d, H_d)$ is*

$$n(\varepsilon, \delta) \leq O\left(\frac{d \log(d) \log(d/\varepsilon) \log(1/\delta)}{\varepsilon}\right).$$

*The query complexity of Comparison-Pool-RPU learning $(D, \mathbb{R}^d, H_d)$ is*

$$q(\varepsilon, \delta) \leq O\left(d \log^2(d/\varepsilon) \log(d) \log(d \log \log(1/\varepsilon)) \log(1/\delta)\right).$$

*Proof.* Recall from Corollary 4.7 that with $n$ samples we can build an RPU learner $A$ with expected coverage:

$$\mathbb{E}[C(A)] \geq \left(1 - \binom{n}{k} g(k)\right) \frac{n - k}{n}.$$

By Theorem 4.10, letting $k = O(d \log(d) \log(n))$ simplifies this to

$$\mathbb{E}[C(A)] \geq \left(1 - \frac{1}{n}\right) \frac{n - k}{n}$$

as long as $n \geq \Omega(\max(c_1 + c_2, k))$. Setting the right hand side to $1 - \varepsilon$ and solving for n gives

$$n = O\left(\frac{d \log(d) \log(d/\varepsilon)}{\varepsilon}\right),$$

and a Chernoff bound gives the desired dependence on $\delta$.

Bounding the query complexity is a bit more nuanced. Since the above analysis requires knowing both comparisons and labels for all sampled points, we cannot simply draw $n(\varepsilon, \delta)$ points and actively learn their labels as we would do in the PAC case. Instead, consider the following algorithm for learning a finite sample $S$ of size $n$ drawn from $D$.

1. Subsample $S' \subset S$, $|S'| = O(d \log(d) \log(n))$.

2. Query labels and comparisons on $S'$.

3. Infer[4] labels in $S$ implied by $Q(S')$.

4. Restrict to the set of uninferred points, and repeat $T = O(\log(n/\varepsilon))$ times.

KLMZ [5] proved that if $S$ has inference dimension at most $O(d \log(d) \log(n))$, with the right choice of constants each round of the above algorithm infers half of the remaining points with probability at least a half. Since we repeat the process $T$ times, a Chernoff bound gives that all of $S$ is learned with probability at least $1 - \varepsilon/3$. Further, notice that if $n$ is sufficiently large:

$$n = O\left(\frac{d \log(d) \log^2(d/\varepsilon)}{\varepsilon}\right),$$

Theorem 4.10 and Observation 4.6 imply $S$ has inference dimension at most $O(d \log(d) \log(n))$ with probability at least $1 - \varepsilon/3$, and further the algorithm queries only a $\varepsilon/3$ fraction of points in the sample.

We argue that any algorithm with such guarantees is sufficient to learn the entire distribution. A variation of this fact is proved in [13], but we repeat the argument here for completeness. Notice that the expected coverage of $A$ over the entire distribution may be rewritten as the probability that $A$ infers some additionally drawn point, that is:

$$\underset{S \sim D^n}{\mathbb{E}}[C(A)] = \underset{x_1, \ldots, x_{n+1} \sim D^{n+1}}{\Pr}[A(x_1, \ldots x_n) \to x_{n+1}]$$

We argue that the righthand side is bounded by the probability that a point is inferred but not queried by $A$ across samples of size $n + 1$. To see this, recall that $A$ operates on $S' = \{x_1, \ldots, x_{n+1}\}$ by querying a set of subsets $S'_1, \ldots, S'_T \subset S'$, where each $S'_{i+1}$ is drawn uniformly at random from points not inferred by $S'_1, \ldots, S'_i$. If $x_{n+1}$ is learned but not queried by $A$, it must be inferred by some subset $S_i$. Such a configuration of subsets is only more likely to occur when running $A(S' \setminus \{x_{n+1}\})$, since the only difference is that at any step where $x_{n+1}$ has not yet been inferred, $A(S')$ might include $x_{n+1}$ in the next sample. Finally, recall that our algorithm infers all of $S'$ in only $\varepsilon|S'|/3$ queries with probability at least $2\varepsilon/3$. Since $x_{n+1}$ is just an arbitrary point from $D$, the probability it is inferred but not queried is then at least $1 - \varepsilon$, which gives the desired coverage.

All that remains is to analyze the query complexity. The total number of queries made is $O(Tk \log(k))$, and repeating this process $\log(1/\delta)$ times returns the desired RPU learner by a Chernoff bound. Thus the total query complexity is:

$$q(\varepsilon, \delta) \leq O\left(d \log^2(d/\varepsilon) \log(d) \log(d \log \log(1/\varepsilon)) \log(1/\delta)\right)$$

$\square$

The necessary conditions in Theorem 4.10 are satisfied by a wide range of distributions. The concentration bound is satisfied by any distribution whose norm has finite expectation, and the anti-concentration bound is satisfied by many continuous distributions. Log-concave distributions, for instance, easily satisfy the conditions.

**Proposition 4.12.** *log-concave distributions satisfy the conditions of Theorem 4.10 with $c_1 = c_2 = O(1)$.*

*Proof.* Any log-concave distribution $D$ is affinely equivalent to an isotropic log-concave distribution $D'$. Isotropic log-concave distributions have the following properties [2]:

1. $\forall a > 0, P_{x \in D'}[||x|| \geq a] \leq e^{-a/\sqrt{d}+1}$

2. All marginals of $D'$ are isotropic log-concave.

3. If $d = 1, P_{x \in D'}[x \in [a, b]] \leq |b - a|$

We want to show that these three properties satisfy the two conditions of Theorem 4.10. Property 1 satisfies condition 1 with constant $c_1 = 1$. Properties 2 and 3 imply condition 2 with constant $c_2 = 2$, as the probability mass of a strip is equivalent to the probability mass of the one dimensional marginal along the normal vector. □

With significant additional work, Balcan and Zhang [3] show that an even more general class of distributions satisfies these properties, $s$-concave distributions.

**Proposition 4.13** (Theorems 5,11 [3])**.** *$s$-concave distributions satisfy the conditions of Theorem 3.10[5] for $s \geq -\frac{1}{2d+3}$ and:*

$$c_1 = \frac{4\sqrt{d}}{c}, c_2 = 4$$

*for some absolute constant $c > 0$.*

Theorem 4.10 provides a randomized comparison LDT for solving the point location problem. Because our method involves reducing to worst case inference dimension, we may use the derandomization technique (Theorem 1.8) of [24] to prove the existence of a deterministic LDT.

**Corollary 4.14** (Restatement of Theorem 1.9)**.** *Let $D$ be a distribution satisfying the criterion of Theorem 1.8, $x \in \mathbb{R}^d$, and $h_1, \ldots, h_n \sim D^n$. Then for $n \geq \Omega(d \log^2(d))$ there exists an LDT using only label and comparison queries solving the point location problem with expected depth*

$$O(d \log(d) \log(d \log(n)) \log^2(n)).$$

# 5    Experimental Results

To confirm our theoretical findings, we have implemented a variant of our reliable learning algorithm for finite samples. For a given sample size or dimension, the query complexity we present is averaged over 500 trials of the algorithm.

## 5.1    Algorithm

We first note a few practical modifications. First, our algorithm labels finite samples drawn from the uniform distribution over the unit ball in $d$-dimensions. Second, to match our methodology in lower bounding Label-Pool-RPU learning, we will draw our classifier uniformly from hyperplanes tangent to the unit ball. Finally, because the true inference dimension of the sample might be small, our algorithm guesses a low potential inference dimension to start, and doubles its guess on each iteration with low coverage.

Our algorithm will reference two sub-routines employed by the original inference dimension algorithm in [5], Query$(Q, S)$, and Infer$(S, C)$. Query$(Q, S)$ simply returns $Q(S)$, the oracle responses to all queries on $S$ of type $Q$. Infer$(S, C)$ builds a linear program from constraints $C$ (solutions to some Query$(Q, S)$), and returns which points in $S$ are inferred.

**Algorithm 2:** Perfect-Learning($N, Q, d, c$)

**Result:** Labels all points in sample $S \sim (B^d)^N$ using query set $Q$

**9** S $\sim (B^d)^N$; Classifier $\sim S^d, B^1$;

**10** Subsample_Size $= d + 1$; Uninferred $= S$; Subsample_List $= []$;

**11 while** *size(Uninferred) $> c \cdot$ size(Subsample_Size)* **do**

**12**      Subsample $\sim$ Uninferred[Subsample_Size];

**13**      Subsample_List.extend(Subsample);

**14**      Inferred_Points $=$ Infer(Uninferred, Query(Q, Subsample_List));

**15**      **if** *size(Inferred_Points) $<$ size(Uninferred)/2* **then**

**16**          Subsample_Size $* = 2$;

**17**      **end**

**18**      Uninferred.remove(Inferred_Points)

**19 end**

**20** Query(Label,Uninferred)

Note that this algorithm is efficient. The while loop runs at most $\log(N)$ times, and each loop solves at most $N$ linear programs with $O(f_Q(N))$ constraints in $d + 1$ dimensions. Thus the total running time of Algorithm 2 is Poly$(N, d)$. Further note for simplicity we have chosen $c = 1$ for labels and $c = 2$ for comparisons and will drop this parameter in the following.

## 5.2 Query Complexity

Our theoretical results state that for an adversarial choice of classifier, the number of queries Perfect-Learning($N$, Comparison, $d$) performs is logarithmic compared to Perfect-Learning($N$, Labels, $d$). The left graph in Figure 3 shows this correspondence for uniformly drawn hyperplanes tangent to the unit ball and sample values ranging from 1 to $2^{10}$ in log-scale. In particular, it is easy to see the exponential difference between the Label query complexity in blue, and the Comparison query complexity in orange. Further, our results suggest that Perfect-Learning($N$, Comparison, $d$) should scale near linearly in dimension. The right graph in Figure 3 confirms that this is true in practice as well.

Figure 3: The left graph shows a log-scale comparison of Perfect-Learning($N$, Label, 3) and Perfect-Learning($N$, Comparison, 3). The right graph shows how Perfect-Learning(256, Comparison, $d$) scales with dimension.

# 6 Further Directions

## 6.1 Average Inference Dimension and Enriched Queries

KLMZ [5] propose looking for a simple set of queries with finite inference dimension $k$ for $d$-dimensional linear separators. In particular, they suggest looking at extending to t-local relative queries, questions which ask comparative questions about $t$ points. Unfortunately, simple generalizations of comparison queries seem to fail, but the problem of analyzing their average inference dimension remains open. When moving from 1-local to 2-local queries, our average inference dimension improved from:

$$2^{-\tilde{O}(n)} \rightarrow 2^{-\tilde{O}(n^2)}$$

If there exist simple relative t-local queries with average inference dimension $2^{-\tilde{O}(n^t)}$ over some distribution $D$, then it would imply a passive RPU-learning algorithm over $D$ with sample complexity

$$n(\varepsilon, \delta) = O\left(\frac{\log\left(\frac{1}{\varepsilon}\right)^{1/(t-1)}}{\varepsilon} \log\left(\frac{1}{\delta}\right)\right)$$

and query complexity

$$q(\varepsilon, \delta) \leq O\left(2f_Q\left(4\log^{1/(t-1)}(n)\right)\log\left(\frac{1}{\delta}\right)\log(n)\right)$$

One such candidate 3-local query given points $x_1, x_2$, and $x_3$ is the question: is $x_1$ closer to $x_2$, or $x_3$? KLMZ suggest looking into this query in particular, and other similar types of relative queries are studied in [37–43].

## 6.2 Average Inference Dimension $\implies$ Lower Bounds

We showed in this paper that average inference dimension provides upper bounds on passive and active RPU-learning, but to show average inference dimension characterizes the distribution dependent model, we would need to show it provides a matching lower bound. The first step in this process would require examining the tightness of our average to worst case reduction.

**Observation 6.1.** *Let* $(D, X, H)$ *have average inference dimension* $\omega(g(k))$. *Then the probability that a random sample* $S \sim D^n$ *has inference dimension* $\leq k$ *is:*

$$1 - g(k)\binom{n}{k} \leq \Pr[\textit{inference dimension of } S \leq k] \leq (1 - g(k))^{n/k}$$

Even with a tight version of Observation 6.1, it is an open problem to apply such a result as a lower bound technique for the PAC or RPU models.

## 6.3 Noisy and Agnostic Learning

The models we have proposed in this paper are unrealistic in the fact that they assume a perfect oracle. RPU-learning in particular must be noiseless due to its zero-error nature. This raises a natural question: *can inference dimension techniques be applied in a noisy or non-realizable setting?* Hopkins, Kane, Lovett, and Mahajan [13] recently made progress in this direction, introducing a relaxed version of RPU-learning called Almost Reliable and Probably Useful learning. They are able to provide learning algorithms under the popular [7, 2, 44–48] Massart [49] and Tsybakov noise [50, 51] models.

However, many problems in this direction remain completely open, such as agnostic or more adversarial settings. It remains unclear whether inference based techniques are robust to these settings, since small adversarial adjustments to the inference LP can cause substantial corruption to its output.

## 6.4   Further Applications of RPU-learning

In this paper we offer the first set of positive results on RPU-learning since the model was introduced by Sloan and Rivest [8]. RPU-learning has potential for both practical and theoretical applications. On the practical side, positive results on RPU-learning, or a slightly relaxed noisy model, may allow us to build predictors with better confidence levels. On the theoretical side efficient RPU-learners have potential applications for circuit lower bounds [52].

## Footnotes

*Department of Computer Science and Engineering, UCSD, California, CA 92092. Email: nmhopkin@eng.ucsd.edu. Supported by NSF Award DGE-1650112

†Department of Computer Science and Engineering / Department of Mathematics, UCSD, California, CA 92092. Email: dakane@eng.ucsd.edu. Supported by NSF CAREER Award ID 1553288 and a Sloan fellowship

‡Department of Computer Science and Engineering, UCSD, California, CA 92092. Email: slovett@cs.ucsd.edu. Supported by NSF CAREER award 1350481, CCF award 1614023 and a Sloan fellowship

[1]Formally, $n(\varepsilon, \delta)$ must also be polynomial in a number of parameters of $C$

[2] We note that in this version of the model, the learner must know the support of the distribution. Since we only use the model for lower bounds, we lose no generality by making this assumption.

[3]This work was later improved to be computationally efficient [32], but no longer achieved optimal query complexity.

[4]As in [5]'s core algorithm, inference is computed via a linear programm with constraints given solely by query responses. No knowledge of minimal-ratio is required.

[5]Condition 1, however, must be changed to $\forall \alpha > 16...$ rather than 0, which does not affect the proof.