[Reviews · NeurIPS 2020]

Review 1

Summary and Contributions: This paper studies active learning with label and comparison queries, extending the works of (Kane et al, 2017, Xu et al, 2017). It gives a new set of fundamental upper and lower bounds on halfspace learning in passive/active, label/label+comparison queries, PAC/RPU(Reliably Probably Useful) models, summarized in Tables 1 and 2 in the paper. - On the lower bound side, it shows that any label-only active learning algorithm must have polynomial query complexity in PAC learning setting, and exponential sample complexity in the RPU learning setting. The main techniques are using existing combinatorial geometry results. - On the upper bound side, it shows that under mild distributional assumptions, label+comparison based active learning algorithms can have logarithmic query complexity in both PAC and RPU learning settings. The main technique is by developing a distribution-dependent variant of inference dimension (Kane et al, 2017) and a novel online-to-batch conversion (Thm 4.11).

Strengths: Overall, I think it is a solid paper that greatly advances the understanding of RPU learning and learning with comparison queries. Specifically: - The development of distribution-dependent inference dimension is an important contribution, which can helps spur more works in distribution-dependent analysis of machine learning (similar to Hanneke's disagreement coefficient / Alexander's capacity function for active and passive learning) - The proof of Theorem 4.10 (distribution-dependent inference dimension for halfspaces) is not trivial, in my understanding. I was initially thinking if some arguments therein can be simplified by using (normalized) VC inequalities, but it does not lead to g(n) = 2^{-Omega(n^1+\alpha)} for some positive alpha, which is crucial for obtaining the results in this paper.

Weaknesses: - Many of the lower bounds seem to be direct applications of existing results of combinatorial geometry (although the tightness of these results are also discussed, and they nicely extend the early negative results of Kivinen on RPU learning)

Correctness: They are correct as far as I have checked, although there are a few subtle points that I need the authors' clarification.

Clarity: While I can understand the statements in this paper, I think the presentation can be much improved. For example: - As RPU learning implies PAC learning, is there really a need to present Algorithm 1 and Theorem 3.5? Aren't we already happy with Theorem 4.11? - in Algorithm 1, Threshold(S) is only informally defined, and an elaboration is needed. I think basically, the algorithm can successfully approximately recover b if it can find two neighboring + and - examples? - In the proof of Proposition 4.2, no high-level idea of the proof was given, which makes it a bit hard to follow (although I agree that it is correct) - In the second half of Corollary 4.7, is the goal only to identify the labels of all examples S drawn? Also, what is the active learning algorithm used here? I am also confused about the statement that "the boosting algorithm in KLMZ [5] is reliable even when when given the wrong inference dimension as input". Are you referring the algorithm in page 16 of that paper (arxiv version)? - In Theorem 4.9, is H here H_{d, \eta}? - In the proof of Theorem 4.10, what is h \times [-\gamma, \gamma]? Is it {x: h(x) \in [-\gamma, \gamma]}? - At the end of the proof of Theorem 4.10, it is said that "\forall h there must exists x, s.t. Q(S_h' - {x}) infers x." Would it be still possible that there exist two hypotheses h_1, h_2 that have _very small minimal-ratio_, and they agree with the queries in Q(S_h' - {x}), but disagree on x? - In Algorithm 2, if g is a constant function, can be just replace it with a constant? To align with the terminology of KLMZ as much as possible, I suggest changing the name "average inference dimension" to e.g.. "inference dimension tail bound". I think people usually use "dimension" to denote values that take integers.

Relation to Prior Work: Yes. It also discusses a subsequent paper that benefits from the techniques in this paper.

Reproducibility: Yes

Additional Feedback: I thank the authors for the reply. My opinion has not changed. But I have follow-up questions that I hope the authors can clarify in the final version: 1. In the second half of Corollary 4.7, in the active RPU learning algorithm, do we need to first compute the value of k that minimizes the right hand side of the equation, then apply KLMZ's algorithm with that value of k? It might be interesting to extend KLMZ's algorithm and analysis to achieve adaptivity to the data-dependent inference dimension. 2. In Theorem 4.10, it would be very helpful to point out that what the inference algorithm used here is (it is still not clear to me if the inference algorithm needs to know the minimal-ratio or not - specifically, in the linear program alluded in page 16 of KLMZ, in combination with Claim 4.10 therein, it seems like the knowledge of minimal-ratio is needed). The algorithm right below Theorem 4.11 should be revised accordingly to incorporate this.


Review 2

Summary and Contributions: This paper studies the power of comparisons in the problem of actively learning (non-homogenous) linear classifiers. There are three main results in the paper: 1) in the PAC learning model, neither active learning nor comparison queries alone provide a significant speed-up over passive learning; 2) in the PUR-learning model, the paper confirms that passive learning with label queries is intractable information-theoretically, and active learning alone provides little improvement; 3) in the PUR-learning model, the comparison oracle provides a significant improvement in both active and passive learning scenarios. In the context of previous work, the techniques of this paper are heavily based on a combination of the inference dimension (Kane, Lovett, Moran, and Zhang [5]) and the (non-efficient version of) margin-based active learning (Balcan and Long [2]). The paper also extends the analysis to the s-concave distribution based on a concentration result in Balcan and Zhang [3].

Strengths: 1. Learning of linear classifiers with both comparison and label oracles were rarely studied in the literature; to the best of my knowledge, existing work includes Kane, Lovett, Moran, and Zhang [5] and Xu, Zhang, Miller, Singh, and Dubrawski [7], but many fundamental questions remain open in the community. Using existing techniques, this paper answers many of these questions. 2. The techniques are novel for the NeurIPS standard. The theoretical analysis seem solid. Experiments are available as a theoretical paper.

Weaknesses: The paper does not consider computational efficiency and noise tolerance (given that the paper is built upon [2], which is a computationally-inefficient algorithm), while the techniques of achieving computational efficiency and noise tolerance in the existing work of active learning are available.

Correctness: The claims and method are correct to me.

Clarity: The paper is well-written and easy to understand.

Relation to Prior Work: The paper did a good job on discussing how this work differs from previous contributions.

Reproducibility: Yes

Additional Feedback: ==========after rebuttal========== I have read the rebuttal, and I am happy to recommend acceptance of this paper.


Review 3

Summary and Contributions: I have reviewed this paper before, the following is an adaptation of my past review. This paper considers the problem of learning non-homogenous linear classifiers using comparison queries over distributions that are weakly concentrated (such as s-concave). The main focus of the paper is on the power of comparison queries for distribution-dependent and Reliable and Probably Useful learning. Comparison queries in addition to the labels reveal which of two instances are closer to the boundary of the classifier. Comparison queries were shown to be useful for improving the query complexity of a learning task. Kane et al.’17 considered distribution-independent setting and showed that under assumptions, such as large margin assumption, comparison queries gain an exponential improvement in query complexity over (label) active learning. The same paper also showed that in some cases, query complexity won’t have an asymptotic improvement over passive or active learning, based on a definition of a notion of “inference dimension”. An example of this is learning halfspaces in a distribution independent setting even in R^3. This paper first shows that learning non-homogenous halfspaces over uniform distribution active or passive-comparison learning individually requires poly(1/eps) queries. Similarly, in the RPU setting passive or active learning individually require (1/eps)^O(d) and comparison passive learning requires (1/eps) and comparison pool based only uses (d polylog(1/eps)).

Strengths: Overall, I like the results of the paper and I think studying comparison and label active learning learning for distribution-dependent settings is very valuable. I also like that the paper studies RPU setting, where the algorithm has know what it’s uncertain about. Here, as mentioned above you need the power of comparison queries to get to d polylog(1/eps) as opposed to 1/eps^d. This model is not as well studied in modern literature, but it’s an important learning model when it comes to robustness learning.

Weaknesses: Don't see a particular weakness.

Correctness: Seems correct.

Clarity: Good

Relation to Prior Work: Yes.

Reproducibility: Yes

Additional Feedback:


Review 4

Summary and Contributions: This paper investigates several scenarios in active learning. Namely, both PAC learning and RPU learning are considered for the model; both pool-based and membership query-based settings are considered for the active learning paradigm; and, both labeling and comparisons are considered for the queries. Several new lower bounds and upper bounds are obtained for the query complexity in those scenarios. Remarkably, it is shown that for comparison queries, a query complexity polynomial in log(1/epsilon) is obtained for both the PAC setting and the RPU setting, under relatively weak assumptions for the distributions. Synthetic experiments are performed for corroborating such results.

Strengths: This is a well-written, well-positioned, and well-motivated paper, with new nontrivial results in active learning. Although the proof for the upper-bound in the PAC setting (Theorem 3.3) is inspired from [2], the upper bound in the RPU setting (Theorems 3.7 & 3.8) relies on advanced techniques in high-dimensional geometry (notably the concept of average inference dimension). In essence, this is a very good theoretical contribution.

Weaknesses: I found no real weaknesses.

Correctness: I am not an expert in high-dimensional geometry, and techniques related to the interesting concept of inference dimension. Yet, as far as I could check, the proofs look correct.

Clarity: As mentioned above, this paper is very well-written: all notations and definitions are clearly presented, and Section 4 is particularly useful for understanding the main concepts and tools used in the proofs.

Relation to Prior Work: To the best of my knowledge, the paper is well-positioned with respect to related work, and clearly explains the main improvements obtained for different scenarios.

Reproducibility: Yes

Additional Feedback: As a minor comment, I would suggest to mention some perspectives of further research, such as active learning with comparison queries in agnostic settings. But a detailed conclusion is already provided in the extended version of the paper.

[Author Response · NeurIPS 2020]

We thank all four reviewers for their encouraging remarks and thorough treatment of our work. We respond individually
to Reviewers 1 and 2, and jointly to Reviewers 3 and 4.

**Reviewer 1 Response:** We thank Reviewer 1 for their in-depth comments and questions on the supplementary materials.
We address questions below. Smaller comment/questions not addressed will still be updated in the final version.

*"Is there really a need to present Alg 1 and Thm 3.5?:"* Alg 1, Thm 3.5, and Thm 3.6 are present in order to show a tight
characterization of the query complexity of Comparison-Pool-PAC learning. While it is true that Theorem 4.11 also
implies an upper bound on the weaker Comparison-Pool-PAC model, it leaves an unnecessary $\log(1/\varepsilon)$ gap between
upper and lower bounds which Theorem 3.6 closes.

*"In Alg 1, Threshold(S) is only informally defined, and an elaboration is needed. I think basically, the algorithm can
successfully approximately recover b if it can find two neighboring + and - examples?"* The procedure Threshold(S)
produces some threshold consistent with labeling $S$ by binary search (note that this procedure may mislabel small
portions of $S$ near the true threshold). We will clarify this in the text. In Theorem 3.6, we show this procedure produces
a good threshold with probability at least $2/3$ based on neighboring $+/-$ examples. This is later amplified by Chernoff.

*"In the second half of Corollary 4.7, is the goal only to identify the labels of all examples S drawn? Also, what is the
active learning algorithm used here?"* Yes, the second half Corollary 4.7 only finds the labels of $S$. The algorithm is
[KLMZ17](Theorem 3.2) (Alg box pg. 16). In the text, we restate this theorem as Theorem 2.1, and reference it in the
proof of Corollary 4.7. We will add further description of [KLMZ17]'s algorithm and why it makes no errors when
assuming the wrong inference dimension (this is because inference dimension controls only the algorithms coverage).

*"In Theorem 4.9, is H here $H_{d,\eta}$? ...What is $h \times [-\gamma, \gamma]$?"* Yes, in Theorem 4.9 $H$ should be $H_{d,\eta}$, and $h \times [-\gamma, \gamma]$ is
$\{x : h(x) \in [-\gamma, \gamma]\}$. We will update these.

*"[is it] possible that there exist two hypotheses $h_1$, $h_2$ that have _very small minimal-ratio_, and they agree with the
queries in $Q(S'_h - \{x\})$, but disagree on x?"* Yes, this is possible for arbitrary $h_1, h_2 \in H_d$. However, this does not
affect our argument, since we have reduced (with very high probability) to the case that $h$ has large minimal ratio with
respect to $S'_h$. Note that this does not cause any errors since it is not an assumption, but rather is based off of a verifiable
structural property of $S$ (no large subset is too close or too far from any $h \in H_d$).

**Reviewer 2 Response:** We thank Reviewer 2 for their comments. While we agree with much of their assessment, we
would like to address two aspects of the review with which we disagree.

**Computational Efficiency:** The Reviewer's only stated weakness is that *"The paper does not consider computational
efficiency and noise tolerance."* While it is true that our work focuses mainly on characterizing the query complexity of
realizable-case learning, the former part of this statement is false: we *do provide computationally efficient algorithms*
for both the RPU and PAC models, and *explicitly state this* in the paper (lines 214, 295). In more detail, our main
contribution, the $\tilde{O}(d \log(1/\varepsilon)^2)$ Comparison-Pool-RPU learning upper bound, is computationally efficient, which also
implies a computationally efficient algorithm for the strictly weaker PAC-model. That said, we realize that this fact
is somewhat hidden in the paper, and thank the reviewer for bringing it to our attention. We will add a discussion of
computational efficiency to the introduction to fix this.

**Novelty and Focus:** We would also like to clarify what we view as a misunderstanding of the focus and novelty of our
work. Reviewer 2 comments mostly on our Comparison-Pool-Pac upper bound (Theorem 3.3) based upon [BL13], and
at one point states *"the paper is built upon [BL13]."* In fact, [BL13] is used only once in the entire paper in order to help
characterize the query complexity of Comparison-Pool-PAC learning. *We would like to highlight that we do not view
this as the works' main contribution or novelty.* Rather, as we state in the paper on lines 217, 251, and 277, the main
novelty and focus of our work lies in the analysis of query and computationally efficient RPU-learning, and especially
in the introduction and development of average inference dimension–a novel tool for analyzing distribution dependent
RPU-learning crucial to these results. It is worth noting that Reviewer 1 asks why we even include the [BL13] based
result given our stronger results on RPU-learning (the only reason is a $\log(1/\varepsilon)$ gap in query complexity between the
two algorithms).

**Reviewers 3 and 4 Response:** We thank Reviewers 3 and 4 for their encouraging comments, and believe both reviews
appropriately frame and summarize our work. In response to Reviewer 4's suggestion, should the paper be accepted we
would be happy to use part of the additional camera-ready page for covering further research perspectives.

**References**
[BL13] Long P. Balcan, M. Active and passive learning of linear separators under log-concave distributions. In
*Proceedings of the 26th Conference on Learning Theory*, 2013.
[KLMZ17] Lovett S. Moran S. Zhang J. Kane, D. Active classification with comparison queries. In *IEEE 58th Annual
Symposium on Foundations of Computer Science*, 2017.


[Meta-Review · NeurIPS 2020]

The reviewer are unanimous in their support of accepting this paper. The paper makes an important contribution to the literature on learning with label queries and comparison queries, showing that comparison queries dramatically improve the query complexity of learning (nonhomogeneous) halfspaces under distribution assumptions, and furthermore the results even hold in the more-challenging "RPU" model (where the predictor must never be wrong, but may abstain with epsilon probability). The approach stems from general principles, and may lead to further follow-up works.